# Non-Stationary Structural Causal Bandits

**Yeahoon Kwon**[1]    **Yesong Choe**[1]    **Soungmin Park**[1]    **Neil Dhir**[2*]    **Sanghack Lee**[1*]

[1]Graduate School of Data Science, Seoul National University    [2]Focused Energy Inc.

{dataofyou, yesong, tjdals0653, sanghack}@snu.ac.kr
neil.dhir@focused-energy.co

## Abstract

We study the problem of sequential decision-making in environments governed by evolving causal mechanisms. Prior work on structural causal bandits—formulations that integrate causal graphs into multi-armed bandit problems to guide intervention selection—has shown that leveraging the causal structure can reduce unnecessary interventions by identifying possibly-optimal minimal intervention sets (POMISs). However, such formulations fall short in dynamic settings where reward distributions may vary over time, due to their static—and thus myopic—nature focuses on immediate rewards and overlooks the long-term effects of interventions. In this work, we propose a non-stationary structural causal bandit framework that leverages temporal structural causal models to capture evolving dynamics over time. We characterize how interventions propagate over time by developing graphical tools and assumptions, which form the basis for identifying non-myopic intervention strategies. Within this framework, we devise POMIS$^+$, which captures the existence of variables that contribute to maximizing both immediate and long-term rewards. Our framework provides a principled way to reason about temporally-aware interventions by explicitly modeling information propagation across time. Empirical results validate the effectiveness of our approach, demonstrating improved performance over myopic baselines.

## 1 Introduction

The Multi-Armed Bandit (MAB) problem is a classic decision-making scenario where an agent sequentially selects actions (arms), each with an unknown but fixed reward distribution, to maximize cumulative reward [Sutton and Barto, 2018]. A common assumption in MAB formulations is that arms are independent, meaning that the reward distribution of one arm does not depend on another [Lai and Robbins, 1985, Auer et al., 2002a]. However, in many real-world settings, actions are not independent—hidden factors may simultaneously influence both the choice of action and the observed reward, introducing unobserved confounders (UCs). Bareinboim et al. [2015], Forney et al. [2017], Lattimore et al. [2016], Zhang and Bareinboim [2017] demonstrate that in the presence of UCs, standard bandit algorithms guarantees—such as convergence to the optimal arm and sublinear regret—no longer hold as observed rewards may not accurately reflect the true effect of actions. To address this, Lee and Bareinboim [2018] use structural causal models (SCMs) [Pearl, 2009] and causal diagrams to identify which variables an agent should intervene on to learn an optimal policy.

In many real-world applications, the reward distribution associated with an action is not fixed but changes over time due to shifting contexts, user behavior or environmental factors. Such settings motivate the study of non-stationary multi-armed bandit (NS-MAB) problems, which extend the classical formulation to handle temporal variation in reward dynamics. This dynamic formulation requires agents to continuously adapt to evolving environments by incorporating temporal information

---

*Corresponding authors

into their decision-making process. In particular, it relaxes the assumption of fixed reward distributions that underpins the classical MAB setting. While prior approaches to NS-MAB address temporal shifts in reward through statistical modeling—such as bounded change models [Auer et al., 2002b] or recharging payoffs [Papadigenopoulos et al., 2022]—they typically model non-stationarity in purely statistical terms, overlooking the underlying causal mechanisms responsible for reward shifts. Yet, in many sequential decision-making problems, such shifts are driven by latent and evolving causal mechanisms. In such cases, ignoring the structural causes behind the reward changes can limit an agent's ability to make informed interventions. In contrast, we frame non-stationarity through the lens of SCMs, allowing us to reason about how interventions propagate over time. The details are provided in Appendix E.

To our knowledge, the non-stationary MAB problem has not yet been studied through the causal lens. As such one could consider adapting existing causal methods for stationary settings. In particular, Lee and Bareinboim [2019] propose a latent projection-based method to identify possibly-optimal arms under partial causal knowledge (as detailed in Appendix J.1). However, that approach does not explicitly account for temporal order, treating the non-stationary process as if it were stationary. As a result, it fails to reveal which arms influence which rewards over time. This matters because, in non-stationary environments, identifying truly optimal strategies—those informed by both causal structure and temporal dynamics—is crucial. It enables more effective and timely interventions in real-world settings such as healthcare, education, and resource allocation, where decisions must often be made under uncertainty and limited budgets. Intervening without regard to temporal structure may appear optimal in the short term but lead to sub-optimal cumulative outcomes, ultimately limiting long-term performance. By explicitly modeling temporal structure, our approach overcomes this by capturing how causal effects unfold over time. This enables non-myopic strategies that align with long-term reward maximization, which is essential for planning in dynamic environments.

In this work, we approach the NS-MAB problem through the causal lens by introducing a structural framework that captures temporal dynamics via a rolled-out causal graph[2] (§3). We show how prior methods like POMIS fail to account for shifting causal structures over time and propose POMIS$^+$ that identifies non-myopic intervention strategies with theoretical guarantees (§5). We provide a graphical formulation of POMIS$^+$ that characterizes how interventions propagate across time, which supports systematic identification of non-myopic optimal interventions (§6). Building on this, we develop an efficient algorithm for computing intervention sequences using these graphical structures (§7). We empirically evaluate POMIS$^+$ across three non-stationary tasks, demonstrating its superiority over the myopic baseline in regret and optimal arm selection by capturing long-term causal effects through temporally-aware interventions (§8). Appendix A outlines the paper's structure.

**Contributions** First, we propose a framework—non-stationary structural causal bandit (NS-SCM-MAB)—that combines the nature of NS-MAB with the underlying SCMs. Our framework models temporal dynamics not only in the reward distributions but also in the underlying causal structure, capturing how interventions propagate over time. This formulation enables temporally-aware decision-making, where an intervention at time $t$ may have effects on both immediate and future outcomes. Second, to support this, we devise POMIS$^+$, which extends the original POMIS by incorporating variables from future time steps to better evaluate the long-term value of interventions. Third, we theoretically establish that a partial assignment to preceding variables can strictly improve the expected reward in the temporal manner. Finally, we design an algorithm that identifies such non-myopic intervention sequences and demonstrates its superiority over the myopic strategy through experiments across a range of non-stationary environments.

## 2 Preliminaries

We introduce notation from causal inference to understand multi-outcome causal MAB problems. Capital letters denote a single variable, and the domain of $X$ is denoted $\mathscr{D}(X)$. Bold capital $\mathbf{X} = \{X_1, \ldots, X_n\}$ represents a set of variables. Additionally, lowercase $x \in \mathscr{D}(X)$ represents a value of variable $X$, and the set of values is described as bold lowercase $\mathbf{x} \in \mathscr{D}(\mathbf{X}) = \times_{X \in \mathbf{X}} \mathscr{D}(X)$. We denote $\mathbf{x}[\mathbf{W}]$ as the values of $\mathbf{x}$ corresponding to the intersection $\mathbf{W} \cap \mathbf{X}$.

---

[2]A rolled-out causal graph is a time-unfolded representation of an SCM that explicitly captures temporal dependencies across time steps. Similar constructions appear in causal modeling over time, see e.g., Koller and Friedman [2009, p. 203].

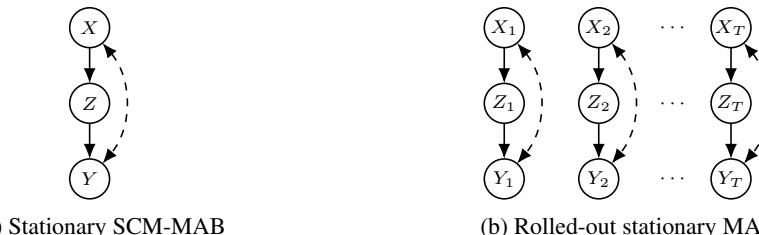

(a) Stationary SCM-MAB         (b) Rolled-out stationary MAB

Figure 1: (a) shows the causal diagram for stationary SCM-MAB. The graph notation for a rolled-out version of the stationary MAB is shown in (b) indexing each round by its corresponding time step.

We adopt the structural causal model (SCM) [Pearl, 2009] framework. An SCM $\mathcal{M}$ is a tuple $\langle \mathbf{U}, \mathbf{V}, \mathcal{F}, P(\mathbf{U}) \rangle$, where $\mathbf{V}$ is a set of endogenous variables and $\mathbf{U}$ is a set of exogenous variables. Each $f_V \in \mathcal{F}$ is a function that determines each endogenous variable $V$. That is, $V \leftarrow f_V(\mathbf{pa}_V, \mathbf{u}_V)$ where $\mathbf{Pa}_V \subseteq \mathbf{V} \setminus \{V\}$ and $\mathbf{U}_V \subseteq \mathbf{U}$. $P(\mathbf{U})$ is a joint distribution over the exogenous variables. The SCM induces a causal diagram $\mathcal{G}$, which includes directed edges encoding functional relationships between endogenous variables, and bi-directed edges encoding UCs. We adopt family relationships $Pa(\cdot)_{\mathcal{G}}$, $Ch(\cdot)_{\mathcal{G}}$, $An(\cdot)_{\mathcal{G}}$, and $De(\cdot)_{\mathcal{G}}$ to denote parents, children, ancestors and descendants of a given variable where ancestors and descendants include its argument.

We consider a discrete-time setting where each time step $t \in \{1, \ldots, T\}$ (abbreviated $t \in [T]$) corresponds to a distinct point in time. We denote by $\mathbf{V}_t \subseteq \mathbf{V}$ a set of variables at each time step $t$, focusing on the specific variables indexed by the subscript $t$. Let $\mathbf{U}_t \subseteq \mathbf{U}$ be the set of exogenous variables, including exogenous variable $\mathbf{U}_{V_t}$ for every $V_t \in \mathbf{V}_t$ and unobserved confounders for every $V_t^i, V_t^j \in \mathbf{V}_t$ if $\mathbf{U}_{V_t^i}$ and $\mathbf{U}_{V_t^j}$ are correlated. $\mathbf{V}_{t<} \subset \mathbf{V}$ is the set of variables whose time index is strictly greater than $t$. Let $\mathbf{X}_t \subseteq \mathbf{V}_t \setminus \{Y_t\}$ denote the set of manipulative variables, where $Y_t$ is the outcome variable. Then, let $\mathbf{N} \subseteq \mathbf{V} \setminus \mathbf{X}_t$ denote the set of non-manipulative variables. We use blackboard bold letter $\mathbb{V} = \{\mathbf{V}_1, \ldots, \mathbf{V}_T\}$ to represent the collection of all time-indexed variables throughout the sequence and use $\mathbb{F} = \{\mathcal{F}_1, \ldots, \mathcal{F}_T\}$ to denote the collection of all time-specific functions.

A vertex-induced subgraph is represented by $\mathcal{G}[\mathbf{V}']$ where $\mathbf{V}' \subseteq \mathbf{V}$. Given a causal diagram $\mathcal{G}$, a time-specific subgraph can be constructed as $\mathcal{G}[\bigcup_{i=t}^{t+\tau-1} \mathbf{V}_i]$, where $t$ denotes the starting time step and $\tau$ represents the length of the time window. In particular, when $\tau = 1$, we refer to this graph as a 'time slice', denoted by $\mathcal{G}[\mathbf{V}_t]$. The probability of $Y = y$, when variables $\mathbf{X}$ are fixed to $\mathbf{x}$, is denoted $P(y \mid \mathrm{do}(\mathbf{X} = \mathbf{x}))$ using the do-operator—an intervened probability. The graphical representation of the intervention is denoted $\mathcal{G}_{\overline{\mathbf{X}}}$; a mutilated graph where the incoming edges onto $\mathbf{X}$ are removed.

## 3 Non-Stationary Multi-Armed Bandit as a Structural Causal Model

We define the NS-SCM-MAB and identify the problems that arise when an existing *stationary* SCM-MAB solution is applied to the NS-MAB problem by reviewing the SCM-MAB [Lee and Bareinboim, 2018]. Let $\mathcal{M} = \langle \mathbf{U}, \mathbf{V}, \mathcal{F}, P(\mathbf{U}) \rangle$ be an SCM, and $Y \in \mathbf{V}$ be a reward variable, where $\mathscr{D}(Y) \subseteq \mathbb{R}$. The arms of the bandits are defined by the possible values $\mathbf{x}$ of the manipulative variable set $\mathbf{X}$, where $\mathbf{X} \subseteq \mathbf{V} \setminus \{Y\}$, and $\mathbf{x} \in \mathscr{D}(\mathbf{X})$. Each arm is associated with a reward distribution $P(Y \mid \mathrm{do}(\mathbf{X} = \mathbf{x}))$. The expected reward of an

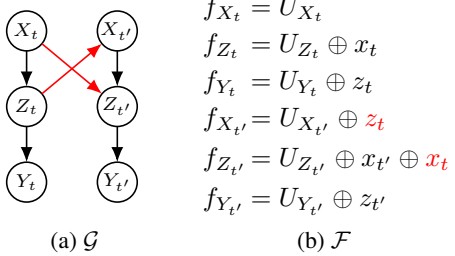

(a) $\mathcal{G}$           (b) $\mathcal{F}$

Figure 2: SCM with non-stationary structure.

arm is denoted by $\mathbb{E}[Y \mid \mathrm{do}(\mathbf{X} = \mathbf{x})]$, written as $\mu_{\mathbf{x}}$. The optimal value of $\mathbf{x}$ for the expected reward is denoted by $\mathbf{x}^*$, i.e., $\mathbf{x}^* = \mathrm{argmax}_{\mathbf{x} \in \mathscr{D}(\mathbf{X})} \mu_{\mathbf{x}}$. For clarity, we represent the information given to an agent interacting with an SCM-MAB as $[\![\mathcal{G}, Y]\!]$. We now discuss time-expanded notation. When we expand it to the time horizon, for every $t \in [T]$, the causal structure among time-indexed variables is identical across all time steps (illustrated in Fig. 1(b)). The arms of the bandits are defined by the possible values $\mathbf{x}_t$ of the set $\mathbf{X}_t \subseteq \mathbf{V}_t \setminus \{Y_t\}$ and $\mathbf{x}_t \in \mathscr{D}(\mathbf{X}_t)$. The reward distribution is

defined $P(Y_t \mid \mathrm{do}(\mathbf{X}_t = \mathbf{x}_t), \mathbb{1}_{t>1} \cdot I_{1:t-1})$ where $I_{1:t-1} = \mathrm{do}(\{\mathbf{X}_i = \mathbf{x}_i\}_{i=1}^{t-1})$ denotes previous interventions and $\mathbb{1}_{t>1}$ is the indicator function. We denote $\mathbb{E}[Y_t \mid \mathrm{do}(\mathbf{X}_t = \mathbf{x}_t), \mathbb{1}_{t>1} \cdot I_{1:t-1}]$ as $\mu_{\mathbf{x}_t, I_{1:t-1}}$, abbreviated as $\mu_{\mathbf{x}_t}$. The optimal value for $\mathbf{X}_t$ is denoted $\mathbf{x}_t^*$.

**Non-stationarity in structural causal models**  Before defining *non-stationarity*, we clarify our understanding of *stationarity*. In the context of bandits, stationarity means that the reward distribution induced from pulling one arm (strategy) among all possible arms remains constant across all time steps $t \in [T]$. Interpreted from the perspective of the SCM-MAB, this implies that the distribution over the outcome variable $Y$—under interventions on any variables in the set of action nodes $\mathbf{X} \subseteq \mathbf{V} \setminus Y$—remains unchanged, regardless of earlier interventions, as discussed in Appendix D. Formally, we can express this as follows: for every pair of time steps $t, t' \in [T]$ with $t \neq t'$, a reward distribution $P(y_t \mid \mathrm{do}(\mathbf{x}_t), \mathbb{1}_{t>1} \cdot I_{1:t-1}) = P(y_{t'} \mid \mathrm{do}(\mathbf{x}_{t'}), \mathbb{1}_{t'>1} \cdot I_{1:t'-1})$. From this point of view, we introduce the NS-SCM-MAB where the reward distribution evolves over time.

**Definition 3.1** (Non-stationary SCM-MAB).  Given a causal diagram $\mathcal{G}$, let $\mathcal{M} = \langle \mathbf{U}, \mathbf{V}, \mathcal{F}, P(\mathbf{U}) \rangle$ be an SCM and $\mathbf{Y} \subset \mathbf{V}$ be a set of reward variables $\mathbf{Y} = \bigcup_{i=0}^T \{Y_i\}$, where $T > 1$. An SCM-MAB $\langle \mathcal{M}, \mathbf{Y} \rangle$ is said to be non-stationary, for some $t, t' \in [T]$ with $t < t'$, if the reward distribution

$$P\left(Y_t \mid \mathrm{do}(\mathbf{X}_t = \mathbf{x}_t), \mathbb{1}_{t>1} \cdot I_{1:t-1}\right) \neq P\left(Y_{t'} \mid \mathrm{do}(\mathbf{X}_{t'} = \mathbf{x}_{t'}), \mathbb{1}_{t'>1} \cdot I_{1:t'-1}\right)$$

The disparity between reward distributions is referred to as a *reward distribution shift*.

The NS-SCM-MAB addresses the problem of arm selection in environments where the interventional distribution over reward variables may shift over time. This is illustrated in Fig. 2. As shown in the diagram, the value from the previous time step ($t$) can influence the subsequent time step ($t'$), as indicated by the red edges. This temporal influence causes the reward distribution to change over time, reflecting the non-stationary nature of the underlying SCM. To understand this setting from the agent's perspective, it is important to recognize that the topological ordering of variable generation is subordinated to the temporal ordering—variables associated with earlier time steps are always instantiated before those at later time steps. Based on this temporal structure, we define the notion of a temporal model within an SCM. A temporal model is an SCM that captures the time-specific causal mechanisms as perceived by the agent at a given time step.

**Definition 3.2** (Temporal Model).  Let $\mathcal{M} = \langle \mathbf{U}, \mathbf{V}, \mathcal{F}, P(\mathbf{U}) \rangle$ be an SCM. For some $t \in [T]$ with $T > 1$, we define values $\mathbf{v}_t^\star$ and $\mathbf{u}_t^\star$, which correspond to the values of $\mathbf{V}_t^\star = Pa(\mathbf{V}_t) \setminus \mathbf{V}_t$ and the set of correlated variables $\mathbf{U}_t^\star$ between $V_t \in \mathbf{V}_t$ and $V_{t'} \in \mathbf{V}_{t'}$ for every $t \neq t'$, respectively. A temporal model $\mathcal{M}_t \mid \mathbf{v}_t^\star, \mathbf{u}_t^\star$ is defined as $\langle \mathbf{U}_t, \mathbf{V}_t, \mathcal{F}_t, P(\mathbf{U}_t \mid \mathbf{u}_t^\star) \rangle$ where each $f_{V_t} \in \mathcal{F}_t$ is a function that determines $V_t$ given predetermined values $\mathbf{v}_t^\star$ and $\mathbf{u}_t^\star$. That is, $V_t \leftarrow f_{V_t}(\mathbf{pa}_{V_t}[\mathbf{V}_t] \cup \mathbf{v}_t^\star, \mathbf{u}_{V_t} \cup \mathbf{u}_t^\star)$ where $\mathbf{Pa}_{V_t} \subseteq \mathbf{V} \setminus \{V_t\}$ and $\mathbf{U}_{V_t} \subseteq \mathbf{U}_t$.

Concisely, we denote $\mathcal{M}_t \mid \mathbf{v}_t^\star, \mathbf{u}_t^\star$ by $\mathcal{M}_t$. Since the agent can *only* manipulate variables within the current time step, the temporal model is regarded as an underlying mechanism from the agent's perspective for each time step. The key point is that the mechanism may vary across time steps, which arises from *information propagation*, resulting in a reward distribution shift. A *myopic* agent, however, fails to adapt to this change and follows a strategy which chooses actions based on the current information only.

**Non-stationarity caused by information propagation**  We introduce the concept of 'information propagation' which induces the reward distribution shift between the two subsequent temporal models, and examine how it arises from the perspective of temporal models. To illustrate this, we consider two temporal models $\mathcal{M}_t = \langle \mathbf{U}_t, \mathbf{V}_t, \mathcal{F}_t, P(\mathbf{U}_t \mid \mathbf{u}_t^\star) \rangle$ and $\mathcal{M}_{t+1} = \langle \mathbf{U}_{t+1}, \mathbf{V}_{t+1}, \mathcal{F}_{t+1}, P(\mathbf{U}_{t+1} \mid \mathbf{u}_{t+1}^\star) \rangle$. To isolate the effect of structural changes, we fix the exogenous distributions by assuming $P(\mathbf{U}_t \mid \mathbf{u}_t^\star) = P(\mathbf{U}_{t+1} \mid \mathbf{u}_{t+1}^\star)$ and $P(\mathbf{u}_t^\star) = P(\mathbf{u}_{t+1}^\star)$. The agent identifies that the data generation for $\mathcal{M}_{t+1}$ occurs only after the data generation governed by $\mathcal{M}_t$. Therefore, the agent at time $t$ can neither observe the values of variables at a future time nor intervene in advance on future variables. Under the environment, $f_{V_t} \in \mathcal{F}_t \neq f_{V_{t+1}} \in \mathcal{F}_{t+1}$ where $\mathcal{F}_t, \mathcal{F}_{t+1} \in \mathbb{F}$ implies that the distribution induced from the SCM-MAB $\langle \mathcal{M}_t, Y_t \rangle$ may differ from the distribution from $\langle \mathcal{M}_{t+1}, Y_{t+1} \rangle$. Such function inequality arises due to predetermined values $\mathbf{v}_{t+1}^\star \in \mathscr{D}(\mathbf{V}_{t+1}^\star)$ where $\mathbf{V}_{t+1}^\star = \mathbf{Pa}_{V_{t+1}} \setminus \mathbf{V}_{t+1}$, which is determined at time step $t$. As a result, information propagation refers to the process by which the predetermined values, generated within a temporal model $\mathcal{M}_t$, influence the generation of a value in another temporal model $\mathcal{M}_{t+1}$.

To understand the concept better, we examine a simple case of two subsequent temporal models depicted in Fig. 2. At time step $t$, the value of $X_t$ is generated from the function $f_{X_t} = U_{X_t}$ and the rest of the values are generated according to the topological (causal) order. At time step $t'$, the function generating $X_{t'}$ is $f_{X_{t'}} = U_{X_{t'}} \oplus z_t$, where $z_t$ is the value of $Z_t$ generated at time step $t$. Thus, at time step $t'$, the function $f_{X_{t'}}$ utilizes $z_t$, treating it as a conditional parameter (i.e., constant at $t'$) to generate the value of $X_{t'}$. In this setting, the reward distribution of the arms may change over time, introducing additional complexity and challenges.

**Graph representation of the NS-SCM-MAB** The graph representation of the NS-SCM-MAB captures the dynamics of the underlying non-stationary structure and the dependencies between variables of different time slices. Each time slice corresponds to a set of variables $\mathbf{V}_t \in \mathbb{V}$ and the relationships between these slices are depicted by edges, which we elaborate on in the sequel, representing the information propagation across time. To establish the conditional independence structure between the causal diagrams of the NS-SCM-MAB and its probability distribution, we assume the *time-slice Markov property* as a temporal extension of the Markov property.

**Assumption 3.1** (Time-slice Markov). Given a causal diagram $\mathcal{G}$ and a probability distribution $P$ relative to $\mathcal{G}$, for every time step $t \in [T]$, if $(\mathbf{V}_t \perp\!\!\!\perp \mathbf{V}_{<t-1} \mid \mathbf{V}_{t-1})_{\mathcal{G}}$ where $\mathbf{V}_t \in \mathbb{V}$, then $P$ is said to be time-slice Markov relative to $\mathcal{G}$.

Assumption 3.1 illustrates that under non-stationary dynamics, the SCM-MAB exhibits a first-order Markovian structure where each time slice depends only on the immediately preceding one.

In addition, the graphical structure and topology of each time slice are identical across all time steps; that is, $\mathcal{G}[\mathbf{V}_t] = \mathcal{G}[\mathbf{V}_{t'}]$ for all $t \neq t' \in [T]$. This reflects the bandit nature of our setting, where each temporal model corresponds to a repeated instance of the same underlying causal structure. Given any two time slices $\mathcal{G}[\mathbf{V}_t]$ and $\mathcal{G}[\mathbf{V}_{t'}]$, this property implies stationarity between the corresponding temporal model MABs $\langle \mathcal{M}_t, Y_t \rangle$ and $\langle \mathcal{M}_{t'}, Y_{t'} \rangle$, provided that no information propagation occurs across time steps, as illustrated in Fig. 1(b).

If there exists at least one edge between $\mathcal{G}[\mathbf{V}_t]$ and $\mathcal{G}[\mathbf{V}_{t'}]$, where $V_t \in \mathbf{V}_t$ and $V_{t'} \in \mathbf{V}_{t'}$, this edge is represented as a *transition edge* $(V_t, V_{t'})$. The edge indicates that information propagation from a temporal model $\mathcal{M}_t$ to another $\mathcal{M}_{t'}$, conforming to the two subsequent time slices $\mathcal{G}[\mathbf{V}_t]$ and $\mathcal{G}[\mathbf{V}_{t'}]$, occurs at every time step. Notably, Assumption 3.1 implies that transition edges may exist only between consecutive time slices (i.e., first-order), thereby prohibiting connections between non-adjacent time slices.

## 4 Motivating Example

The primary challenge we address is identifying action *sequences* that maximize the expected reward under the NS-SCM-MAB setting. We first contemplate the solution of the stationary SCM-MAB problem given a causal diagram $\mathcal{G}$ to understand the non-stationary causal bandit. In the stationary MAB setting, choosing a set of arms can be interpreted as identifying a set of nodes $\mathbf{Z}$ from $\mathbf{V} \setminus \{Y\}$ that maximize the expected reward for outcome variable $Y$, formally written as

$$\mathbf{z}^\dagger \triangleq \underset{\mathbf{z} \in \mathscr{D}(\mathbf{Z}), \mathbf{Z} \subseteq \mathbf{V} \setminus \{Y\}}{\arg\max} \mathbb{E}[Y \mid \mathrm{do}(\mathbf{Z} = \mathbf{z})]. \tag{1}$$

Previously, Lee and Bareinboim [2018] proposed a *do-calculus*-based approach to exclude intervention sets that may induce sub-optimal expected rewards, ultimately identifying possibly optimal minimal intervention sets (POMISs) [Lee and Bareinboim, 2018, Def. 2]. We demonstrate, however, that this approach may itself lead to sub-optimality in the NS-SCM-MAB setting.

In the non-stationary setting, unlike the stationary setting where the agent only computes a POMISs once, the agent explores the search space at every time step to identify the POMISs for each time-indexed reward variable. For instance, in Fig. 3(a), we may consider the process by which an agent selects an intervention set at each time step. We can identify the POMISs by utilizing the concepts of Minimal UC-Territory (MUCT) [Lee and Bareinboim, 2018, Def. 3] and Interventional Border (IB) [Lee and Bareinboim, 2018, Def. 4], which characterize the POMISs graphically.

At time step $t = 1$, from the agent's perspective, the set of possible exploration sets is $\{\emptyset, \{X_1\}, \{Z_1\}, \{X_1, Z_1\}\}$. We use an algorithm enumerating all POMISs [Lee and Bareinboim,

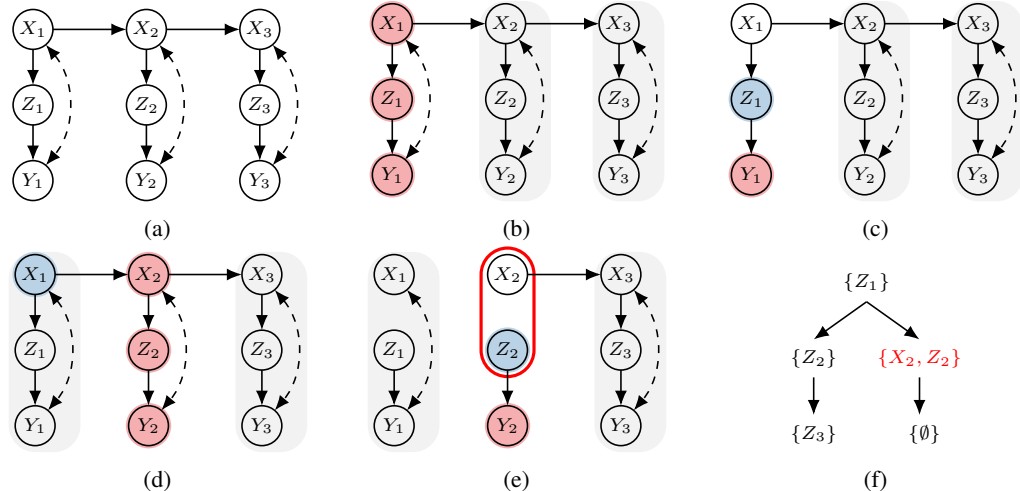

Figure 3: (a) Non-stationary causal diagram $\mathcal{G}$ with $t \in \{1,2,3\}$. (b-d) illustrate examples of MUCT/IB based on the agent's movement at each time step. (e) represents the agent's non-myopic intervention strategy at $t = 2$ (red box). (f) depicts a tree diagram of all possible scenarios when action selection starts with $\{Z_1\}$ (red text denotes POMIS$^+$).

2018, Alg. 1] with respect to $[\![\mathcal{G}[\mathbf{V}_1], Y_1]\!]$. The algorithm generates $\text{IB}(\mathcal{G}[\mathbf{V}_1], Y_1) = \emptyset$ and $\text{IB}(\mathcal{G}[\mathbf{V}_1]_{\overline{X_1}}, Y_1) = \{Z_1\}$ from $\text{MUCT}(\mathcal{G}[\mathbf{V}_1], Y_1) = \{X_1, Z_1, Y_1\}$ and $\text{MUCT}(\mathcal{G}[\mathbf{V}_1]_{\overline{X_1}}, Y_1) = \{Y_1\}$. These are illustrated in Fig. 3(b) and Fig. 3(c), with the MUCT shaded in red and the IB shaded in blue. Thus, we get POMISs with respect to $[\![\mathcal{G}[\mathbf{V}_1], Y_1]\!]$—i.e., POMIS$_1 = \{\emptyset, \{Z_1\}\}$. As a result, following the same procedure at each time step $t = 1, 2, 3$, the agent obtains the following POMISs:

$$\text{POMISs}_1 = \{\emptyset, \{Z_1\}\}, \text{POMISs}_2 = \{\emptyset, \{Z_2\}\}, \text{POMISs}_3 = \{\emptyset, \{Z_3\}\}$$

Up to these time steps, the agent has identified the intervention set within its observable range at each time step. We next examine whether the identified intervention sets remain valid over an extended time horizon under the non-stationary setting. As illustrated with shaded blue in Fig. 3(d), $\text{MUCT}(\mathcal{G}[\mathbf{V}_1 \cup \mathbf{V}_2], Y_2)$ induces $\text{IB}(\mathcal{G}[\mathbf{V}_1 \cup \mathbf{V}_2], Y_2) = \{X_1\}$. Consequently, under the broader causal diagram $\mathcal{G}[\mathbf{V}_1 \cup \mathbf{V}_2]$, we deduce that POMIS$_2 = \{\{X_1\}, \{Z_2\}\}$. By applying the same procedure to $[\![\mathcal{G}[\mathbf{V}_1 \cup \mathbf{V}_2 \cup \mathbf{V}_3], Y_3]\!]$, we obtain POMIS$_3 = \{\{X_2\}, \{Z_3\}\}$. Eventually, we obtain the following POMISs with a broader horizon:

$$\text{POMISs}_1 = \{\emptyset, \{Z_1\}\}, \text{POMISs}_2 = \{\{X_1\}, \{Z_2\}\}, \text{POMISs}_3 = \{\{X_2\}, \{Z_3\}\}$$

This differs from the previously computed myopic evaluations. These discrepancies highlight that variables such as $\{X_1\}$ or $\{X_2\}$, excluded under myopic evaluations, may be optimal when viewed from a broader temporal perspective—while seemingly valid options like $\emptyset$ become sub-optimal.

Therefore, the agent should have intervened on $\{X_1\}$ and $\{X_2\}$ in advance because those variables are not manipulable for each current time step; $\{X_1\}$ for $t = 2$ and $\{X_2\}$ for $t = 3$. However, the variables may not have been considered optimal for a reward variable at an advanced time step. This is supported by the inequality below:

$$\max_{x_2 \in \mathscr{D}(X_2)} \mu_{x_2} \leq \max_{z_2 \in \mathscr{D}(Z_2)} \mu_{z_2} \tag{2}$$

which is derived from $\mu_{x_2^*} = \sum_{z_2} \mu_{z_2} P(z_2 | \text{do}(x_2^*)) \leq \sum_{z_2} \mu_{z_2^*} P(z_2 | \text{do}(x_2^*)) = \mu_{z_2^*}$ given $[\![\mathcal{G}[\mathbf{V}_2], Y_2]\!]$. This inequality shows that the expected reward for intervening on $\{X_2\}$ is strictly dominated by that of $\{Z_2\}$, making $\{X_2\}$ a suboptimal intervention for $Y_2$. Nonetheless, this does not imply that $\{X_2\}$ should be excluded from the intervention set. Instead, it suggests that $\{X_2\}$ and $\{Z_2\}$ must be jointly intervened upon (i.e., $\{X_2, Z_2\}$ in Fig. 3(e)): while $\{Z_2\}$ is required to optimize the immediate reward $Y_2$, $\{X_2\}$ is necessary to influence the future reward $Y_3$. Importantly, intervening on $\{X_2\}$ does not block the causal influence of $\{Z_2\}$ on $Y_2$, thereby allowing both short-term and long-term rewards to be considered simultaneously. Similarly, for $t = 1$, the

intervention set is assembled as $\{X_1, Z_1\}$. Finally, we obtain the set of intervention sequences: $\{(\{Z_1\}, \{Z_2\}, \{Z_3\}), (\{Z_1\}, \{X_2, Z_2\}, \emptyset), (\emptyset, \{Z_2\}, \{Z_3\}), (\emptyset, \{Z_2\}, \{X_3, Z_3\}), (\emptyset, \{X_2, Z_2\}, \emptyset), (\{X_1, Z_1\}, \emptyset, \{Z_3\})\}$—the details of which are formalized in Def. 5.1. We can depict the tree structure of these sequences as shown in Fig. 3(f), where each branch illustrates a valid path if $Z_1$ is intervened at the first time. These combinations will be discussed in detail in §5.

Building on the above example, we formally define the concept of POMIS for the NS-SCM-MAB and propose a graphical method to capture its dynamic evolution while conserving optimality.

## 5 Conservation of Optimality

We examine theoretical guarantees for identifying optimal intervention sets under the NS-SCM-MAB problem. From §4, we know that myopic intervention (i.e., pulling arms within the current manipulable variables) does not always guarantee optimality for a broader horizon. Rather, the agent must consider the sequence in which variables should be intervened on for subsequent time steps—a *non-myopic* strategy. To formalize this notion, we define an intervention sequence as follows.

**Definition 5.1** (Intervention Sequence). Given $[\![\mathcal{G}, \mathbf{Y}]\!]$, let $\mathbb{S} = (\mathbf{X}_1, \ldots, \mathbf{X}_T)$ be an intervention sequence where $\mathbf{X}_t \subseteq \mathbf{V}_t \setminus \{Y_t\}$ for $t \in [T]$.

Now, consider intervention sequence $(\{X_1, Z_1\}, \emptyset, \{Z_3\})$ in Fig. 3. In the first intervention set of the sequence, $\{X_1, Z_1\}$, Eq. (3) shows that $X_1 \in \mathbf{V}_1$ may serve as a predetermined value for the temporal model $\mathcal{M}_2$, potentially contributing to the maximization of the total reward. For a model $\mathcal{M}$, the maximized sum of expected rewards over two time steps $t = 1, 2$ can be divided into two sums of maximized expected rewards, each corresponding to the temporal model $\mathcal{M}_1$ and $\mathcal{M}_2$

$$\mathbb{E}^{\mathcal{M}}[Y_1 + Y_2 \mid \mathrm{do}(\mathbf{X} = \mathbf{x})] = \mathbb{E}^{\mathcal{M}_1}[Y_1 \mid \mathrm{do}(\mathbf{Z} = \mathbf{z})] + \mathbb{E}^{\mathcal{M}_2 | \mathbf{v}_2^\star = \{x_1\}}[Y_2 \mid \mathrm{do}(\mathbf{K} = \mathbf{k})]. \quad (3)$$

In this equation, we define $\mathbf{X} = \{X_1, Z_1, \emptyset\}$, $\mathbf{Z} = \{Z_1\}$ and $\mathbf{K} = \emptyset$. The value $x_1$ is a predetermined value of $\mathcal{M}_2$ that is taken from the do-assignment at time $t = 1$. From this example, Thm. 5.1 shows that there exists an optimal temporal model under the specific pre-determined values.

**Theorem 5.1.** *(Existence of Optimal Partial Assignment). Given information $[\![\mathcal{G}, \mathbf{Y}]\!]$, let $\mathbf{X}_{t'}$ be a POMIS with respect to $[\![\mathcal{G}[\mathbf{V}_{t'}], Y_{t'}]\!]$ for the subsequent two time steps $t, t' \in [T]$ with $t < t'$. Then, there exists an assignment $\mathbf{w}^*$ for a subset of variables $\mathbf{W} \subseteq Pa(\mathbf{V}_{t'}) = \mathbf{V}_{t'}^\star$ such that*

$$\mathbb{E}^{\mathcal{M}_{t'} | \mathbf{v}_{t'}^\star}[Y_{t'} \mid \mathrm{do}(\mathbf{x}_{t'}^*)]$$

*achieves the maximum expected reward for any $\mathbf{v}_{t'}^\star$ with $\mathbf{w}^* = \mathbf{v}_{t'}^\star[\mathbf{W}]$.*

*Remark* 5.1. The subset $\mathbf{W} \subseteq Pa(\mathbf{V}_{t'})$ in Thm. 5.1, when combined with the intervention set $\mathbf{X}_{t'}$, can be interpreted as a POMIS with respect to $[\![\mathcal{G}[\mathbf{V}_t \cup \mathbf{V}_{t'}], Y_{t'}]\!]$. In other words, although $\mathbf{W}$ lies in time step $t$ and $\mathbf{X}_{t'}$ in time step $t'$, their union forms a possibly-optimal intervention set for $Y_{t'}$ under the expanded temporal context, highlighting the intertemporal nature of optimal control.

Theorem 5.1 implies that once the optimal partial assignment $\mathbf{w}^*$ is fixed, the remaining variables in $Pa(\mathbf{V}_{t'}) \setminus \mathbf{W}$ have no influence on the expected reward of $Y_{t'}$. Therefore, the agent must select these variables with the original POMIS at $t$.

**Definition 5.2** (POMIS with Future Support (POMIS$^+$)). Given information $[\![\mathcal{G}, \mathbf{Y}]\!]$, for subsequent time step $t, t' \in [T]$ with $t < t'$, let $\mathbf{X}_t$ be a POMIS with respect to $[\![\mathcal{G}[\mathbf{V}_t], Y_t]\!]$. If there exists $\mathbf{W}_t \subseteq \mathbf{V}_t \setminus \{Y_t\}$ satisfying Thm. 5.1 such that $\mu_{\mathbf{x}_t} = \mu_{\mathbf{x}_t \cup \mathbf{w}_t}$ for every temporal model $\mathcal{M}_t$ conforming to $\mathcal{G}[\mathbf{V}_t]$, then $(\mathbf{X}_t \cup \mathbf{W}_t)$ is called POMIS with future support for $Y_{t'}$, denoted POMIS$^+_{t,t'}$.

As the name suggests, POMIS$^+$ extends the original POMIS by incorporating one additional $\mathbf{W}_t$. This set consists of variables that contribute to the maximization of the expected reward at a subsequent time step. However, the selection of $\mathbf{W}_t$ is constrained by the temporal structure of causal diagram $\mathcal{G}$.

**Proposition 5.1.** *(Temporal Dependency). Given causal diagram $\mathcal{G}$ and a collection of time-specific variables $\mathbb{V}$, POMIS$^+_{t,t+1} \subseteq \mathbf{V}_t \setminus \{Y_t\}$ for every $t \in [T]$ under Assumption 3.1.*

Proposition 5.1 implies that in causal diagram $\mathcal{G}$, we can determine POMIS$^+$ by only exploring the causal diagram window $\mathcal{G}[\mathbf{V}_t \cup \mathbf{V}_{t+1}]$.

---

**Algorithm 1** Computing all intervention sequences

---

1: **function** POMIS⁺($\mathcal{G}$, $\mathbb{V}$, $\mathbb{Y}$, $T$)
2: **Input**: $\mathbb{G}$ a causal diagram; $\mathbb{V}$ a collection of time indexed variables;
       $\mathbb{Y}$ a collection of time indexed rewards; $T$ the time horizon of an episode
3: **return** POMIS⁺SEQ ($\mathcal{G}$, $\mathbb{V}$, $\mathbb{Y}$, $T$)

4: **function** POMIS⁺SEQ($\mathcal{G}$, $\mathbb{V}$, $\mathbb{Y}$, $T$, $[\mathcal{I}^+ = \emptyset]$, $[\mathcal{Q} = \emptyset]$)
5:   $\mathbb{S}, Y_t, \mathcal{G}' \leftarrow \emptyset, \mathbb{Y}[T], \mathcal{G}[An(Y_t)_{\mathcal{G}}]$
6:   $\mathbb{P} \leftarrow \{(\text{MUCT}(\mathcal{G}'_{\overline{\mathbf{W}}}, Y_t), \text{IB}(\mathcal{G}'_{\overline{\mathbf{W}}}, Y_t))\}_{\mathbf{W} \subseteq An(Y_t)_{\mathcal{G}} \setminus \{Y_t\}}$
7:   **for** (Xs, Ts) $\in \mathbb{P}$ **do**
8:     $\mathcal{I}^+, \mathcal{Q} \leftarrow \text{IB}^+(\mathbb{V}, \text{Xs}, \mathcal{I}^+), \text{QIB}(\mathcal{G}, \mathbb{V}, \text{Ts}, \mathcal{Q}, \mathcal{I}^+)$ (by Alg. 3)
9:     $t_e \leftarrow$ the earliest time index $t$ such that $t \in \text{keys}(\mathcal{I}^+)$
10:    **if** $t_e > 1$ **then** $\mathbb{S} \leftarrow \mathbb{S} \cup (\text{POMIS}^+\text{SEQ}(\mathcal{G}, \mathbb{V}, \mathbb{Y}, t_e-1, \mathcal{I}^+, \mathcal{Q}))$
11:    **else** $\mathbb{S} \leftarrow$ the product combination of all sets $\{\mathcal{I}^+[t] \cup q_t \mid q_t \in \mathcal{Q}[t]\}_{t \in \text{keys}(\mathcal{I}^+) \cap \text{keys}(\mathcal{Q})}$ (by Prop. 6.1)
12: **return** $\mathbb{S}$

---

# 6 Graphical Characterization of POMIS⁺

To graphically characterize POMIS⁺, we introduce two novel graphical concepts—interventional borders over future rewards (IB⁺) and qualified interventional borders (QIB)—which together enable the construction of POMIS⁺ based on causal topological properties.

**Definition 6.1** (IB for the Subsequent Time Steps (IB⁺)). Given $[\![\mathcal{G}, \mathbf{Y}]\!]$, for any $\mathbf{W} \subseteq An(Y_{t'})_{\mathcal{G}} \setminus \{Y_{t'}\}$, define $\mathbf{Z}(\mathbf{W}) = \text{IB}(\mathcal{G}[\bigcup_{i=t}^{t'} \mathbf{V}_i]_{\overline{\mathbf{W}}}, Y_{t'})$. Then, for each $j \in \{t, \ldots, t'\}$, $\mathbf{Z}(\mathbf{W}) \cap \mathbf{V}_j$ is called an Interventional Border at time slice $j$ for $Y_{t'}$, denoted by $\text{IB}_{j,t'}^+$.

Intuitively, $\text{IB}_{t,t'}^+$ is computed for the future reward $Y_{t'}$ but resides within $\mathbf{V}_t$. Therefore, we must identify a *qualified* interventional border under $[\![\mathcal{G}[\mathbf{V}_t], Y_t]\!]$—that is, an IB that satisfies certain conditions to remain effective even when $\text{IB}_{t,t'}^+$ is fixed.

**Definition 6.2** (Qualified IB (QIB)). Let $\text{IB}_{t,t'}^+(\mathcal{G}[\bigcup_{i=t}^{t'} \mathbf{V}_i], Y_{t'}) = \mathbf{X}^+$ where $t, t' \in [T]$ with $t < t'$. If $\text{IB}(\mathcal{G}[\mathbf{V}_t]_{\overline{\mathbf{X}^+ \cup \mathbf{W}}}, Y_t) = \mathbf{X}$ for every $\mathbf{W} \subseteq \mathbf{V}_t \setminus \{Y_t\}$, then $\mathbf{X}$ is a Qualified Interventional Border for a reward variable $Y_t$, denoted $\text{QIB}_t$.

According to the definition, the QIB is selected such that the variables in $\text{IB}_{t,t'}^+$, though fixed to optimize $Y_{t'}$, do not block the causal influence on the current reward $Y_t$. By taking the union of both $\text{IB}_{t,t'}^+$ and $\text{QIB}_t$, we can identify $\text{POMIS}_{t,t'}^+$.

**Proposition 6.1.** *(Composition of POMIS⁺).* Given $[\![\mathcal{G}, \mathbf{Y}]\!]$, $\text{IB}_{t,t'}^+(\mathcal{G}[\bigcup_{i=t}^{t'} \mathbf{V}_i], Y_{t'}) \cup \text{QIB}_t(\mathcal{G}[\mathbf{V}_t], Y_t)$ *is a* $\text{POMIS}_{t,t'}^+$ *for* $t, t' \in [T]$ *and* $t < t'$.

As shown in Fig. 4, an IB for $Y_{t'}$ spans over two time steps (blue shaded), and $\{X_t\}$ is an $\text{IB}_{t,t'}^+$ from Def. 6.1. The candidates for $\text{QIB}_t$ w.r.t. $[\![\mathcal{G}[\mathbf{V}_t], Y_t]\!]$ are $\{Z_t\}$ and $\{W_t\}$. If $\{W_t\}$ is chosen for $\text{QIB}_t$, then one candidate for $\text{POMIS}_{t,t'}^+$ is $\{X_t, W_t\}$ (light gray), whereas if the choice is $\{Z_t\}$, then $\text{POMIS}_{t,t'}^+$ is $\{X_t, Z_t\}$ (dark gray). However, by Def. 6.2, we choose $\{Z_t\}$ for $\text{QIB}_t$ (purple shaded) not $\{W_t\}$, which can be explained using do-calculus: we can show that $\mu_{x_t^*, w_t^*} = \mu_{x_t^*} \leq \mu_{z_t^*} = \mu_{x_t^*, z_t^*}$. This inequality suggests that $\{X_t, Z_t\}$ is determined as $\text{POMIS}_{t,t'}^+$ rather than $\{X_t, W_t\}$. Note that, under Assumption 3.1, it is sufficient to consider only a subsequent time step to obtain POMIS⁺ without examining other time steps. In other words, given $[\![\mathcal{G}, \mathbf{Y}]\!]$, the union of $\text{IB}_{t,t+1}^+(\mathcal{G}[\mathbf{V}_t \cup \mathbf{V}_{t+1}], Y_{t+1})$ and $\text{QIB}_t(\mathcal{G}[\mathbf{V}_t], Y_t)$ constitutes a $\text{POMIS}_{t,t+1}^+$.

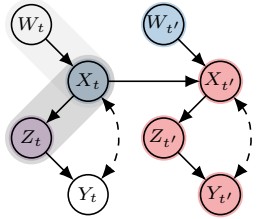

Figure 4: POMIS for the subsequent time step.

# 7 Algorithmic Characterization

We propose a recursive algorithm (Alg. 1) for constructing optimal intervention strategies under non-stationarity, which systematically explores the graph in a backward manner while avoiding exhaustive

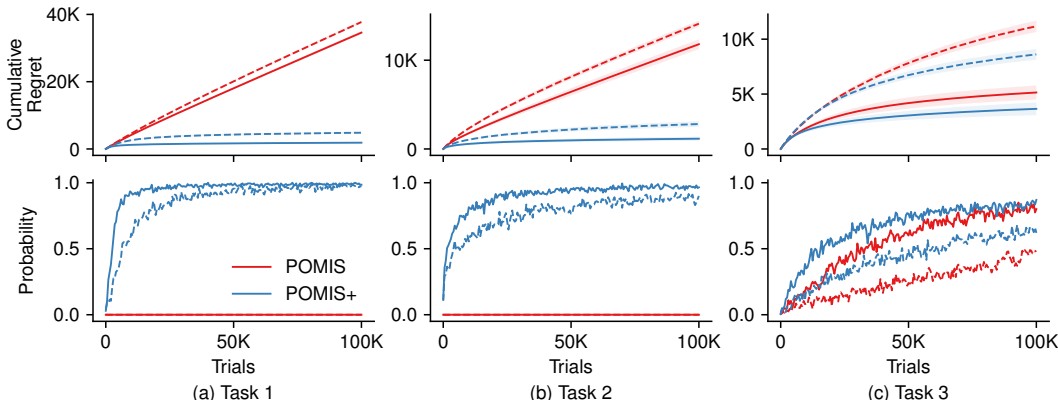

Figure 5: Cumulative regret (top) and optimal arm selection probabilities (bottom) across tasks (columns), with solid for Thompson Sampling and dashed for KL-UCB; shaded bands denote $\pm 1$ SD.

enumeration of all variable subsets. Given a causal diagram $\mathcal{G}$ and a sequence of reward variables $\mathbb{Y}$, the algorithm first constructs a subgraph $\mathcal{G}'$ consisting of ancestors of the target reward variable $Y_t$. Then, all valid MUCT and IB pairs are identified in the graph $\mathcal{G}'$(Line 6). The algorithm iteratively explores each MUCT/IB pair, updating two key maps: $\mathcal{I}^+$ and $\mathcal{Q}$, corresponding to IB$^+$ (Def. 6.1) and QIB (Def. 6.2), respectively (Line 8). Both $\mathcal{I}^+$ and $\mathcal{Q}$ map each time $t$ to the selected intervention variables via the functions IB$^+$ and QIB. These maps are propagated through the recursive structure of the algorithm, gradually expanding the intervention variables at each time step. If the earliest time-step of variables currently stored in IB$^+$ has not yet reached start time ($t = 1$), the algorithm recursively calls itself to evaluate earlier time steps, taking the current $\mathcal{I}^+$ and $\mathcal{Q}$ as arguments (Line 10). Conversely, suppose IB$^+$ reaches the start time, indicating completion of the backward enumeration. In this case, the algorithm generates all possible intervention combinations for each time step based on the final IB$^+$ and QIB maps, as stated in Prop. 6.1. It then enumerates all possible intervention sequences by taking the Cartesian product of these timestep-specific combinations, adding each sequence to the intervention sequence container $\mathbb{S}$ (Line 11). The construction of POMIS$^+$ sequences leverages graphical characteristics (i.e., IB$^+$ and QIB) to reduce redundant computations across time. The time complexity is exponential in both the horizon $T$ and the number of variables $|\mathbf{V}|$, specifically $\mathcal{O}(2^{2|\mathbf{V}|\cdot T})$. The details of the time complexity are available in Appendix H. We now show that the recursive enumeration performed by the algorithm is exhaustive and correct with respect to the definition of POMIS$^+$ sequences.

**Theorem 7.1.** *(Soundness and Completeness). Given information $[\![\mathcal{G}, \mathbf{Y}]\!]$, Alg. 1 returns all intervention sequences composed solely of POMIS$^+$ sets.*

## 8 Experiments

We evaluate how effectively the POMIS$^+$ strategy captures long-term effects compared to the myopic POMIS baseline. We conduct experiments[3] on three settings, each designed to highlight different aspects of temporal intervention planning. Detailed specifications for each task are provided in App. I. We report two metrics—cumulative regret (CR) and optimal arm selection probability (OAP)—under two MAB solvers: Thompson Sampling (TS) [Thompson, 1933] and KL-UCB [Cappé et al., 2013].

**Task 1** (Fig. 3(a)): As shown in Fig. 5(a), CR of the TS for the POMIS$^+$ strategy converges around step 20K (solid blue line). In contrast, the sequences constructed from the myopic POMIS strategy show no sign of convergence (solid red line). Similar patterns are observed for the OAP result. Quantitatively, by step 100K, the POMIS$^+$ strategy achieves a CR of 1.9K$\pm$29 (TS) and 4.8K$\pm$43 (KL-UCB), with optimal-arm probabilities (OAP) of 98.5$\pm$1.7% and 97.0$\pm$2.4%, respectively. Meanwhile, the POMIS baseline performs substantially worse, with CR values exceeding 34K and 37K, while OAP scores remain near zero. This result is attributed to the fact that myopic strategies fail to include intervention nodes such as $X_1$ for MUCT($\mathcal{G}[\mathbf{V}_2], Y_2$) and $X_2$ for MUCT($\mathcal{G}[\mathbf{V}_3], Y_3$). The inability to access such variables leads to critical long-term failures in non-stationary settings.

---

[3]A Python implementation can be found at: `https://github.com/yeahoon-k/NS-SCMMAB`.

**Task 2** (Fig. 4): As shown in the second column of Fig. 5, the POMIS$^+$ strategy consistently outperforms the myopic POMIS baseline in both cumulative regret and optimal arm selection probability. By step 100K, POMIS$^+$ achieves substantially lower regrets—1.1K$\pm$32 (TS) and 2.8K$\pm$44 (KL-UCB)—compared to 11.8K$\pm$77 and 14.1K$\pm$48 for the POMIS strategy. Furthermore, the OAP of POMIS$^+$ reaches 96.5$\pm$2.6% (TS) and 90.0$\pm$4.2% (KL-UCB), while the POMIS baseline remains near zero. These results demonstrate that POMIS$^+$ successfully captures the long-term effect of $X_t$ on $Y_{t'}$, which is overlooked by the myopic strategy.

**Task 3** (Fig. 6): The final experiment is based on the SCM, which intentionally violates the time-slice Markov assumption (Assumption 3.1) by introducing long-range dependencies and latent interactions across time (Appendix F). As shown in Fig. 5(c), POMIS$^+$ outperforms the baseline, though the gap is smaller than in previous tasks. By step 100K, the CR of POMIS$^+$ is 3.7K$\pm$80 (TS) and 8.6K$\pm$69 (KL-UCB), improving upon the POMIS baseline regret of 5.1K$\pm$89 and 11.2K$\pm$72, respectively. In terms of OAP, POMIS$^+$ achieves 87.0$\pm$4.7% (TS) and 62.0$\pm$6.7% (KL-UCB), while POMIS reaches only 80.0$\pm$5.6% and 48.5$\pm$6.9%, respectively. These results highlight that although the advantage of POMIS$^+$ is par-

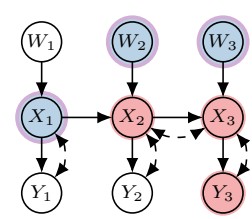

Figure 6: $\mathcal{G}$ for Task 3.

tially reduced due to the absence of time-local independence, it still provides superior performance by leveraging earlier interventions that propagate across time via DUCs.

## 9 Discussion

We assume access to the full causal structure during evaluation. In many real-world settings, however, the agent may have only limited or partial access to future causal diagrams rather than complete knowledge of them. Extending our framework to handle partially observed or uncertain temporal structures—while maintaining optimality—remains an important direction for future work.

While our proposed algorithm is theoretically sound and complete, its computational complexity is constrained by the exponential growth of the search space. A promising direction is to exploit repeated or modular structures that frequently appear in non-stationary environments—for instance, by developing segment-wise variants of our algorithm that reuse computed intervention sequences across recurring causal patterns. Such extensions could preserve theoretical soundness while substantially improving scalability and applicability to larger, real-world domains.

For the experimental setting, each experimental trial corresponds to a complete causal rollout, where outcomes from all time steps are aggregated into a single episode-level reward. This episodic formulation enables fair comparison across intervention sequences based on their overall causal influence over time. Although interventions are not executed sequentially across evolving environments, the causal dependencies among time-indexed variables are fully encoded in the temporal models, ensuring valid evaluation of the structural optimality of POMIS$^+$ sequences. Further discussions are provided in Appendix K.

## 10 Conclusion

In this work, we present a novel framework; the NS-SCM-MAB, that integrates non-stationary multi-armed bandit problems with structural causal models. Our formulation captures both evolving reward distributions and shifting causal structures, enabling temporally-aware decision-making that accounts for how interventions propagate over time. To formalize this, we introduce POMIS$^+$, a temporally-extended notion of optimal intervention sets, and provide theoretical justification showing that conditioning on preceding variables can achieve improved expected rewards in dynamic environments. Building on this, we develop an algorithm to identify non-myopic intervention strategies, and demonstrate its empirical advantages over myopic baselines across a variety of non-stationary settings. We believe this work provides a foundation for causal reasoning in sequential decision-making under temporal dynamics and opens up new avenues for research at the intersection of causal inference and non-stationary environments.

## Acknowledgments and Disclosure of Funding

We thank anonymous reviewers for constructive comments to improve the manuscript. This work was partly supported by the IITP (RS-2022-II220953/25%, RS-2025-02263754/25%) and NRF (RS-2023-00211904/25%, RS-2023-00222663/25%) grants funded by the Korean government. Yesong Choe was supported in part by Basic Science Research Program through the NRF funded by the Ministry of Education (RS-2025-25418030).

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

# A  Organization

A schematic of our framework and its logical flow is illustrated in Fig. 7. The diagram visually maps out how the paper develops, starting from the formulation of the non-stationary SCM-MAB problem in §3, and introducing the temporal model along with key graphical assumptions about the nature of non-stationarity. §5 establishes that in a temporal model, partial assignments of predetermined variables can affect optimal interventions at future time steps, motivating the definition of POMIS$^+$ as a non-myopic intervention. This leads into the graphical characterization in §6, which defines key structures (IB$^+$, QIB) that construct POMIS$^+$. Finally, §7 presents an algorithm that enumerates optimal intervention sequences based on these graphical elements. The structure clarifies the dependencies between the paper's components and highlights how our contributions build upon one another.

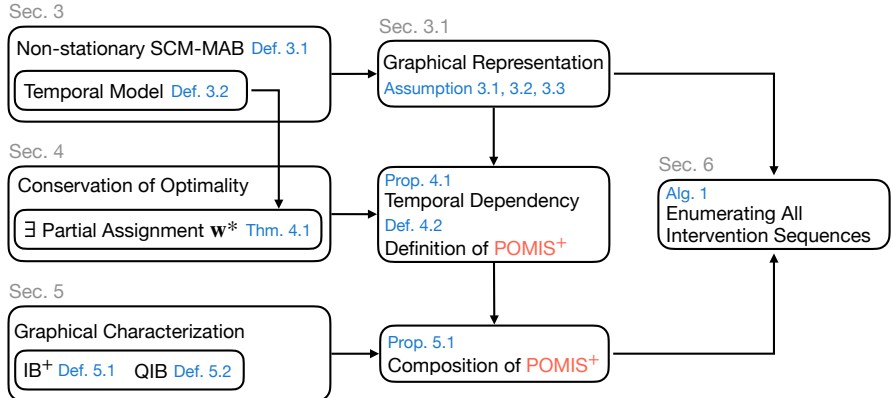

Figure 7: An overall schema of this paper.

# B  Preliminaries

Here we present definitions of MIS, POMIS, MUCT, and IB.

**Definition B.1.** (Minimal Intervention Set (MIS) [Lee and Bareinboim, 2018]). A set of variables $\mathbf{X} \subseteq \mathbf{V} \setminus \{Y\}$ is said to be a *minimal intervention set* relative to $[\mathcal{G}, Y]$ if there is no $\mathbf{X}' \subset \mathbf{X}$ such that $\mu_{\mathbf{x}[\mathbf{X}']} = \mu_{\mathbf{x}}$ for every SCM conforming to the $\mathcal{G}$.

**Definition B.2.** (Possibly-Optimal Minimal Intervention Set (POMIS) [Lee and Bareinboim, 2018]). Given information $[\mathcal{G}, Y]$, let $\mathbf{X}$ be a MIS. If there exists an SCM conforming to $\mathcal{G}$ such that $\mu_{\mathbf{x}^*} > \forall_{\mathbf{Z} \in \mathbb{Z} \setminus \{\mathbf{X}\}} \mu_{\mathbf{z}^*}$, where $\mathbb{Z}$ is the set of MISs with respect to $\mathcal{G}$ and $Y$, then $\mathbf{X}$ is a *possibly-optimal minimal intervention set* with respect to the information $[\mathcal{G}, Y]$.

**Definition B.3.** ($C$-component [Tian and Pearl, 2003]). In a causal diagram $\mathcal{G}$, two variables are said to be in the same confounded component (for short, $C$-component or $\mathsf{CC}(\cdot)_{\mathcal{G}}$ if and only if they are connected by a bi-directed edge (i.e., a path composed solely of $V_i \leftrightarrow V_j$).

In this paper, we denote $\mathsf{CC}(\mathbf{X})_{\mathcal{G}} = \bigcup_{X \in \mathbf{X}} \mathsf{CC}(X)_{\mathcal{G}}$.

**Definition B.4.** (Unobserved-Confounders' Territory [Lee and Bareinboim, 2018]). Given information $[\mathcal{G}, Y]$, let $H$ be $\mathcal{G}[An(Y)_{\mathcal{G}}]$. A set of variables $\mathbf{T} \subseteq \mathbf{V}(H)$ containing $Y$, where $\mathbf{V}(H)$ is the set of variables in $H$, is called a *UC-territory* on $\mathcal{G}$ with respect to $Y$ if $\mathrm{De}(\mathbf{T})_H = \mathbf{T}$ and $\mathsf{CC}(\mathbf{T})_H = \mathbf{T}$.

A UC-territory $\mathbf{T}$ is said to be *minimal* if no $\mathbf{T}' \subset \mathbf{T}$ is a UC-territory. A minimal UC-Territory (MUCT) for $\mathcal{G}$ and $Y$ can be constructed by extending a set of variables, starting from $\{Y\}$, alternatively updating the set with the c-component and descendants of the set.

**Definition B.5.** (Interventional Border [Lee and Bareinboim, 2018]). Let $\mathbf{T}$ be a minimal UC-territory on $\mathcal{G}$ with respect to $Y$. Then, $\mathbf{X} = pa(\mathbf{T})_{\mathcal{G}} \setminus \mathbf{T}$ is called an *interventional border* for $\mathcal{G}$ with respect to $Y$.

---
**Algorithm 2** Minimal unobserved confounders' territory
---
1: **function** MUCT($\mathcal{G}$, $\mathbb{Y}$)
2: $H = \mathcal{G}[An(Y)_\mathcal{G}]$
3: $\mathbf{Q} = \{Y\}$; $\mathbf{T} = \{Y\}$
4: **while** $\mathbf{Q} \neq \emptyset$ **do**
5:    remove an element $Q_1$ from $\mathbf{Q}$
6:    $\mathbf{W} = \mathsf{CC}(Q_1)_H$; $\mathbf{T} = \mathbf{T} \cup \mathbf{W}$; $\mathbf{Q} = (\mathbf{Q} \cup de(\mathbf{W})_H) \setminus \mathbf{T}$
7: **return** $\mathbf{T}$
---

## C  Nomenclature

| | |
|---|---|
| NS | Non-stationary |
| SCM | Structural causal model |
| RL | Reinforcement learning |
| MAB | Multi-armed bandit |
| MIS | Minimal intervention set |
| UC | Unobserved confounder |
| POMIS | Possibly-optimal minimal intervention set |
| MUCT | Minimal unobserved confounders' territory |
| IB | Interventional border |
| QIB | Qualified interventional border |
| DUC | Dynamically unobserved confounder |
| CR | Cumulative regret |
| OAP | Optimal arm-selection probability |
| KL-UCB | Kullback-Leibler upper confidence bound |

## D  Stationary SCM-MAB

The cumulative regret of the stationary setting is given by:

$$R_T \triangleq \sum_{t=1}^{T} \left( \mu_{\mathbf{x}^\dagger} - \mathbb{E}\left[\mu_{\mathbf{x}_t}\right] \right), \tag{4}$$

where $\mathbf{x}^\dagger$ denotes the globally optimal arm, defined as

$$\mathbf{x}^\dagger \triangleq \operatorname*{arg\,max}_{\mathbf{x} \in \mathscr{D}(\mathbf{X}),\, \mathbf{X} \subseteq \mathbf{V} \setminus \{Y\}} \mathbb{E}[Y \mid \mathrm{do}(\mathbf{X} = \mathbf{x})].$$

where $\mu_{\mathbf{x}_t}$ is the arm played at round $t$. The SCM-MAB setting assumes that the agent has full access to the causal graph $\mathcal{G}$ of $\mathcal{M}$, although its parametrization remains unknown—i.e., the agent knows the structure $\mathcal{G}$, but not the structural functions $\mathcal{F}$ or the distribution over exogenous variables $P(\mathbf{U})$. Furthermore, the causal graph $\mathcal{G}$ is assumed to be static, meaning that the underlying causal structure of the domain does not change over time. As a result, the agent interacts with the same causal model in each round.

In this setting, there is no confounding across time slices, and thus, no information propagates between rounds. Consequently, the reward distribution remains fixed across rounds.

In practice, under the stationary setting, the agent effectively observes, in each round, a causal diagram $\mathcal{G}$ composed of temporally disconnected slices, as illustrated in Fig. 1(b). This stationary (and also non-stationary) formulation typically assumes that the number of arms is constant throughout the interaction. The graphical structure and topology of each time slice are identical across all rounds.

The SCM-MAB framework inherently introduces dependencies between arms, stemming from the underlying causal relationships among endogenous and exogenous variables. Lee and Bareinboim [2018, 2019] identified two structural properties that can be derived from any SCM-MAB framework:

1. Arm equivalence: a characterization of arms that share identical reward distributions, determined using constraints from do-calculus, and

2. Partial-orders among arms: under what topological conditions one arm can be optimal.

Leveraging these properties, one can identify minimal intervention sets (MIS) that constitute a non-redundant collection of informative interventions. In addition, Lee and Bareinboim [2019] identified both MIS and POMIS for the stationary setting with non-manipulative variables. However, these characterizations rest on the assumption that the causal graph $\mathcal{G}$ remains static and stationary— meaning that no information is carried over from previous decisions. §3 extends beyond the stationary assumption to present the SCM-MAB framework in the non-stationary setting.

## E  Comparison with Conventional Non-Stationary Bandit Algorithms

We contrast our approach to non-stationarity in the SCM-MAB with traditional non-stationary bandit settings. Our focus is on how causal modeling provides a structured explanation for reward distribution shifts over time, as opposed to treating them as statistical artifacts. These build upon early foundational work on dynamic allocation and index policies [Whittle, 1988, Gittins, 1979].

**Conventional NS-bandit formulations**  Conventional non-stationary bandit algorithms aim to maintain *low regret* with respect to a comparator (or competitor) class—a predefined set of benchmark policies that may adapt over time to account for changing environments (e.g., policies that allow a limited number of switches between arms, comparators that track the best-performing arm over recent time windows, or strategies that assume bounded changes in the underlying reward distribution). These algorithms are typically categorized into two regimes: adversarial and stochastic bandits, illustrated in Table 1.

Table 1: Representative non-stationary bandit settings and algorithms (adapted from Lattimore and Szepesvári [2020]).

| Regime | Setting | Description | Representative Algorithms |
|---|---|---|---|
| Adversarial | $L$-switching | The identity of the optimal arm may change abruptly up to $L$ times | Exp3.S, AdaHedge |
| Adversarial | Variation budget | Total variation in reward sequences is bounded by $V$ | Rexp3, Adapt-EvE |
| Stochastic | Piecewise-stationary | Rewards are stationary within intervals, with occasional change-points | Sliding-window UCB, Change-point Thompson Sampling |
| Stochastic | Drifting | Expected rewards evolve smoothly over time | Discounted UCB, Sliding-window UCB, SW-TS |

Each setting assumes non-stationarity as a statistical property: either abrupt shifts, or slowly drifting rewards. Some algorithms require knowledge of the number of switches $L$, while others are designed to be adaptive. In the variation budget setting, the cumulative amount of change in reward distributions is bounded by a budget $V$, offering a finer-grained control of non-stationarity than simple change-point models.

**Causal non-stationarity (our perspective)**  Our framework models non-stationarity as a consequence of causal information propagation across time. Specifically, transition edges in the causal graph $\mathcal{G}$ (e.g., $X_t \rightarrow Z_{t'}$) induce changes in the downstream reward variables (e.g. $Y_{t'}$), illustrated in Fig. 2. This structure directly captures *why* the reward distribution changes.

For instance, whereas traditional settings might treat $\mathbb{E}[Y_t \mid do(x)]$ as shifting arbitrarily with $t$, our approach identifies structural causes: $P(Y_t \mid do(X_t))$ is influenced by information propagation

from previous variables (e.g. $X_{t-1}$, $Z_{t-1}$). This allows us to model the *mechanism* behind reward distribution shifts.

Moreover, the presence of arcs (or edges) between time slices in $\mathcal{G}$ determines where and how changes occur. This is closely aligned with the "mean payoff drift" interpretation in traditional models [Lattimore and Szepesvári, 2020, Chapter 31], but we interpret it in terms of explicit graphical information in $\mathcal{G}$.

**Summary** Most traditional algorithms detect or adapt to change, but do not explain it. Our approach, by contrast, offers a mechanism-based explanation of non-stationarity—one grounded in a causal understanding of the system under investigation, which is itself a strong assumption—via the SCM structure. This allows for:

- Identification of reward-relevant intervention targets (POMIS$^+$)
- Intervention sequence planning backed by theoretical guarantees derived from the underlying causal structure

Ultimately, our method treats non-stationarity as a structured phenomenon emergent from a dynamic causal model, rather than as an arbitrary change in observed statistics.

# F    Semi Time-Slice Markovian Non-Stationary SCM-MAB

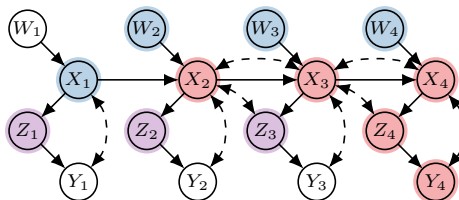

Figure 8: IB$^+$ ranges over four time steps.

We begin by formalizing dynamic unobserved confounders (DUCs), which represent exogenous variables inducing dependencies across time.

**Definition F.1.** (Dynamic unobserved confounders (DUCs)). Given $[\![\mathcal{G}, \mathbf{Y}]\!]$, let $\mathbf{U}_t^\star$ denote the set of exogenous variables that induce dependencies between variables in $\mathbf{V}_t$ and $\mathbf{V}_{t'}$ for some $t < t'$. If $U_j \in \mathbf{U}_t^\star$, then we refer to $U_j$ as a dynamic unobserved confounder (DUC), indicating that $U_j$ introduces confounding effects that persist across time.

When such DUCs are permitted in the graphical structure, we say the temporal graph satisfies the semi-time-slice Markov property.

**Definition F.2** (Semi-time-slice Markov). A temporal graph $\mathcal{G}$ satisfies the semi-time-slice Markov property if it allows the presence of dynamic unobserved confounders (DUCs) across time. That is, the graph permits bidirected edges between variables $V_t \in \mathbf{V}_t$ and $V_{t'} \in \mathbf{V}_{t'}$ for $t \neq t'$, representing confounding induced by exogenous variables $U_j \in \mathbf{U}_t^\star$ that simultaneously affect both time slices.

Under the semi-time-slice Markov assumption induced by dynamic unobserved confounders (Def. F.1), we need to perform a time-expanded search for valid intervention targets across multiple time steps. Consider the example in Fig. 8. When we calculate IB$^+$ for MUCT$(\mathcal{G}[\bigcup_{i=1}^4 \mathbf{V}_i], Y_4)$, the result consists of four sets: IB$_{1,4}^+ = \{X_1\}$, IB$_{2,4}^+ = \{W_2\}$, IB$_{3,4}^+ = \{W_3\}$ and IB$_{4,4}^+ = \{W_4\}$ (blue shaded). Given those IB$^+$ at each time step, we can determine each QIB$_t$ according to Def. 6.2. For example, when IB$_{1,4}^+ = \{X_1\}$ is given, the IB corresponding to the MUCT$(\mathcal{G}[\mathbf{V}_1]_{\overline{X_1 \cup Z_1}}, Y_1) = \{Y_1\}$ is $Z_1$, hence QIB$_1 = \{Z_1\}$. Similarly, for MUCT$(\mathcal{G}[\mathbf{V}_2]_{\overline{X_2 \cup W_2}}, Y_2)$, we can select QIB$_2 = \{Z_2\}$, and for MUCT$(\mathcal{G}[\mathbf{V}_3]_{\overline{X_3 \cup W_3}}, Y_3)$, we can choose QIB$_3 = \{Z_3\}$ (purple shaded). For now, by Prop. 6.1, we can obtain POMIS$^+$s by taking the union of each IB$_{t,t'}^+$ and QIB$_t$—POMIS$_{1,4}^+ = \{X_1, Z_1\}$, POMIS$_{2,4}^+ = \{W_2, Z_2\}$, POMIS$_{3,4}^+ = \{W_3, Z_3\}$ and POMIS$_{4,4}^+ = \{W_4\}$. These POMIS$^+$ sets can then be used as components of an intervention sequence in the NS-SCM-MAB setting.

# G Proofs

**Theorem 5.1.** *(Existence of Optimal Partial Assignment). Given information $[\![\mathcal{G}, \mathbf{Y}]\!]$, let $\mathbf{X}_{t'}$ be a POMIS with respect to $[\![\mathcal{G}[\mathbf{V}_{t'}], Y_{t'}]\!]$ for the subsequent two time steps $t, t' \in [T]$ with $t < t'$. Then, there exists an assignment $\mathbf{w}^*$ for a subset of variables $\mathbf{W} \subseteq Pa(\mathbf{V}_{t'}) = \mathbf{V}_{t'}^{\star}$ such that*

$$\mathbb{E}^{\mathcal{M}_{t'}|\mathbf{v}_{t'}^{\star}}[Y_{t'} \mid \mathrm{do}(\mathbf{x}_{t'}^*)]$$

*achieves the maximum expected reward for any $\mathbf{v}_{t'}^{\star}$ with $\mathbf{w}^* = \mathbf{v}_{t'}^{\star}[\mathbf{W}]$.*

*Proof.* Let $t, t' \in [T]$ with $t < t'$ be two time steps. Given the information $[\![\mathcal{G}, \mathbf{Y}]\!]$, we fix an arbitrary target variable $Y_{t'} \in \mathbf{Y}$ located in the time slice $\mathcal{G}[\mathbf{V}_{t'}]$.

We now consider any temporal model $\mathcal{M}_{t'}$ that conforms to the time slice $\mathcal{G}[\mathbf{V}_{t'}]$. Let $\mathbf{X}_{t'}$ be a POMIS for $Y_{t'}$ with respect to the time slice $\mathcal{G}[\mathbf{V}_{t'}]$. The goal is to show that under any temporal model conforming to $\mathcal{G}[\mathbf{V}_{t'}]$, there exists a partial assignment $\mathbf{w}^*$ that preserves the optimality of $\mathbf{X}_{t'}$.

By the definition of POMIS, there exists an intervention assignment $\mathbf{x}_{t'}^* \in \mathscr{D}(\mathbf{X}_{t'})$ such that:

$$\mathbb{E}^{\mathcal{M}_{t'}}[Y_{t'} \mid \mathrm{do}(\mathbf{x}_{t'}^*)] > \mathbb{E}^{\mathcal{M}_{t'}}[Y_{t'} \mid \mathrm{do}(\mathbf{z})] \quad \text{for all } \mathbf{Z} \in \mathbb{Z} \setminus \{\mathbf{X}_{t'}\},$$

where $\mathbb{Z}$ is the set of all MISs for $Y_{t'}$ in $\mathcal{G}[\mathbf{V}_{t'}]$.

Now consider how expectations are evaluated in the temporal model $\mathcal{M}_{t'}$: they are conditioned on predetermined variables $\mathbf{v}_{t'}^{\star} \in \mathscr{D}(Pa(\mathbf{V}_{t'}))$. Hence, there must exist at least one such $\mathbf{v}_{t'}^{\star}$ under which the above inequality holds.

Fix any such $\mathbf{v}_{t'}^{\star}$. Since the domain $\mathscr{D}(Pa(\mathbf{V}_{t'}))$ is finite, and the conditional expectation $\mathbb{E}^{\mathcal{M}_{t'}|\mathbf{v}_{t'}^{\star}}[Y_{t'} \mid \mathrm{do}(\cdot)]$ is a deterministic function of $\mathbf{v}_{t'}^{\star}$ and the do-intervention, there exists a minimal subset $\mathbf{W} \subseteq Pa(\mathbf{V}_{t'})$ such that $\mathbf{w}^* = \mathbf{v}_{t'}^{\star}[\mathbf{W}]$ satisfies:

$$\mathbb{E}^{\mathcal{M}_{t'}|\mathbf{v}_{t'}'}[Y_{t'} \mid \mathrm{do}(\mathbf{x}_{t'}^*)] > \mathbb{E}^{\mathcal{M}_{t'}|\mathbf{v}_{t'}'}[Y_{t'} \mid \mathrm{do}(\mathbf{z})], \quad \text{for all } \mathbf{v}_{t'}' \text{ such that } \mathbf{v}_{t'}'[\mathbf{W}] = \mathbf{w}^*.$$

That is, fixing the partial assignment $\mathbf{w}^*$ suffices to preserve the optimality of the POMIS $\mathbf{X}_{t'}$ regardless of how the remaining variables in $Pa(\mathbf{V}_{t'}) \setminus \mathbf{W}$ are instantiated.

Since the choice of $\mathcal{M}_{t'}$ was arbitrary (subject to conforming to $\mathcal{G}[\mathbf{V}_{t'}]$), this concludes that such a subset $\mathbf{W}$ and partial assignment $\mathbf{w}^*$ must exist under any temporal model consistent with the time-slice causal structure. $\qquad \square$

**Proposition 5.1.** *(Temporal Dependency). Given causal diagram $\mathcal{G}$ and a collection of time-specific variables $\mathbb{V}$, $\mathrm{POMIS}_{t,t+1}^{+} \subseteq \mathbf{V}_t \setminus \{Y_t\}$ for every $t \in [T]$ under Assumption 3.1.*

*Proof.* For the sake of contradiction, suppose that the proposition does not hold. Then, there exists some $t \in [T]$ such that:

$$\nexists \mathrm{POMIS}_{t,t+1}^{+} \subseteq \mathbf{V}_t \setminus \{Y_t\}$$

That is, there exists a POMIS$^+$ set $\mathbf{X}_{t,t+1}^{+}$ such that:

$$\exists X \in \mathbf{X}_{t,t+1}^{+} \text{ where } X \notin \mathbf{V}_t \setminus \{Y_t\}$$

By this existence, $X$ must either lie in a future time slice ($X \in \mathbf{V}_{t'}$ with $t < t'$), or $X = Y_t$.

We now argue that such an $X$ cannot be part of any valid POMIS$^+$ set under Assumption 3.1. Recall that under the assumption, the only causal influences on $Y_{t+1}$ must flow through variables in $\mathbf{V}_t$ (i.e., no time cycles and no backward edges from $\mathbf{V}_{t'}$ with $t < t'$).

Furthermore, $\mathrm{POMIS}_{t,t+1}^{+}$ is defined as the minimal set in $\mathbf{V}_t$ such that the intervention at time $t$ maximizes the expected reward at time $t + 1$. Including any $X \notin \mathbf{V}_t \setminus \{Y_t\}$ violates both this minimality and the validity of the do-intervention within time $t$.

This leads to a contradiction. Hence, for all $t \in [T]$, we must have:

$$\mathrm{POMIS}_{t,t+1}^{+} \subseteq \mathbf{V}_t \setminus \{Y_t\}$$

$\qquad \square$

We now prove the composition property of $POMIS^+_{t,t'}$, which guarantees that interventions on $IB^+_{t,t'}$ do not block the causal effect of $QIB_t$ on $Y_t$. Before proceeding, we formalize the fact that the set constructed by $IB^+_{t,t'}$ satisfies the partial assignment condition required by Thm. 5.1.

**Proposition G.1** ($IB^+$ satisfies the condition of Thm. 5.1). *Let $t < t'$ and let $\mathbf{X}_{t'}$ be a POMIS for $Y_{t'}$ in $\mathcal{G}[\mathbf{V}_{t'}]$. Then, the set $IB^+_{t,t'}(\mathcal{G}[\bigcup_{i=t}^{t'} \mathbf{V}_i], Y_{t'})$ identifies a subset of $Pa(\mathbf{V}_{t'})$ such that there exists a partial assignment $\mathbf{w}^*$ over this set which satisfies the condition of Thm. 5.1.*

*Proof.* Let $t < t'$ and let $\mathbf{X}_{t'}$ be a POMIS for $Y_{t'}$ in $\mathcal{G}[\mathbf{V}_{t'}]$.

From Thm. 5.1, this implies the existence of a subset $\mathbf{W} \subseteq Pa(\mathbf{V}_{t'})$ and a partial assignment $\mathbf{w}^*$ such that for all $\mathbf{v}^\star_{t'}$ with $\mathbf{v}^\star_{t'}[\mathbf{W}] = \mathbf{w}^*$, the reward under $do(\mathbf{x}^*_{t'})$ can be maximized.

Now consider the construction of $IB^+_{t,t'}(\mathcal{G}[\bigcup_{i=t}^{t'} \mathbf{V}_i], Y_{t'})$. By definition, this set is formed by computing the interventional border $IB$ in the full unrolled graph $\mathcal{G}$, and selecting from it only those variables that reside in $\mathbf{V}_t$.

Since the $IB$ of $Y_{t'}$ is known to be a POMIS, and since $IB^+$ is a subset of this $IB$ restricted to variables available at time $t$, the values assigned to $IB^+$ in any temporal model appear as part of some $\mathbf{v}^\star_{t'} \in \mathscr{D}(Pa(\mathbf{V}_{t'}))$.

Note that although Thm. 5.1 is stated in terms of a temporal model $\mathcal{M}_{t'}$, the structure of $\mathcal{M}_{t'}$—including its structural functions and the set of predetermined variables $Pa(\mathbf{V}_{t'})$—is dependent on the $\mathcal{M}$. Since $IB^+_{t,t'}$ is computed from the global graph $\mathcal{G}[\bigcup_{i=t}^{t'} \mathbf{V}_i]$, it selects variables that reflect this SCM-induced structure, and thus corresponds to the $\mathbf{W}$ required in Thm. 5.1.

Therefore, $IB^+_{t,t'}$ fulfills the role of $\mathbf{W}$ in Thm. 5.1, ensuring that fixing its values preserves the optimality of $\mathbf{X}_{t'}$ consistent with $\mathbf{w}^*$. $\qquad\square$

**Proposition 6.1.** *(Composition of $POMIS^+$).* Given $[\![\mathcal{G}, \mathbf{Y}]\!]$, $IB^+_{t,t'}(\mathcal{G}[\bigcup_{i=t}^{t'} \mathbf{V}_i], Y_{t'}) \cup QIB_t(\mathcal{G}[\mathbf{V}_t], Y_t)$ *is a* $POMIS^+_{t,t'}$ *for* $t, t' \in [T]$ *and* $t < t'$.

*Proof.* Let $\mathbf{X}_t := QIB_t(\mathcal{G}[\mathbf{V}_t], Y_t)$ and $\mathbf{W}_t := IB^+_{t,t'}(\mathcal{G}[\bigcup_{i=t}^{t'} \mathbf{V}_i], Y_{t'})$. By Def. 6.2, $\mathbf{X}_t$ is a Qualified Interventional Border (QIB) for $Y_t$ with respect to $\mathcal{G}[\mathbf{V}_t]$, which implies that $\mathbf{X}_t$ is a POMIS for $Y_t$. Thus, the first condition of Def. 5.2 is satisfied.

Next, as established in Prop. G.1, the set $\mathbf{W}_t$ constructed via $IB^+_{t,t'}$ satisfies the condition of Thm. 5.1: there exists a partial assignment $\mathbf{w}_t$ over $\mathbf{W}_t$ such that the expected reward for $Y_{t'}$ under $do(\mathbf{x}_{t'})$ is maximized.

Moreover, due to the construction of $QIB_t$, which requires that $IB(\mathcal{G}[\mathbf{V}_t]_{\overline{\mathbf{Z} \cup \mathbf{W}_t}}, Y_t) = \mathbf{X}_t$ for some $\mathbf{Z} \subseteq \mathbf{V}_t \setminus \{Y_t\}$, it follows that interventions on $\mathbf{W}_t$ do not interfere with the causal effect of $\mathbf{X}_t$ on $Y_t$ within $\mathcal{G}[\mathbf{V}_t]$.

Therefore, the expected reward at time $t$ remains unchanged $\mu_{\mathbf{x}_t} = \mu_{\mathbf{x}_t \cup \mathbf{w}_t}$. Hence, by Def. 5.2, the combined set $\mathbf{X}_t \cup \mathbf{W}_t$ is a $POMIS^+_{t,t'}$. $\qquad\square$

**Theorem 7.1.** *(Soundness and Completeness). Given information $[\![\mathcal{G}, \mathbf{Y}]\!]$, Alg. 1 returns all intervention sequences composed solely of $POMIS^+$ sets.*

*Proof.* We prove the theorem by showing (i) soundness: every sequence returned by Alg. 1 is composed solely of $POMIS^+$, and (ii) completeness: every valid sequence of $POMIS^+$ is returned.

**(Soundness)** Fix the final time step $T$. Let $\mathcal{G}' = \mathcal{G}[An(Y_T)_\mathcal{G}]$ and suppose the algorithm selects a valid MUCT/IB pair $(\mathbf{X}, \mathbf{T})$ under the $\mathcal{G}'$. $\mathsf{IB}^+(\mathbb{V}, \mathbf{X}, \mathcal{I}^+)$ updates the $IB^+$ map by assigning each $X \in \mathbf{X}$ to its respective time step $t_X$. For each $t \in \mathtt{keys}(\mathcal{I}^+)$, the function $\mathsf{QIB}(\mathcal{G}, \mathbb{V}, \mathbf{T}, \mathcal{Q}, \mathcal{I}^+)$ constructs a mutilated graph $\mathcal{G}[\mathbf{V}_t]_{\overline{\mathcal{I}^+[t]}}$, and from it computes all valid POMISs (i.e., $\mathsf{q}_t$) that do not intersect with $\mathbf{T}$. In the base case (i.e., when the earliest $t_e = 0$), the Cartesian product over all $t$ gives sequences $\{\mathcal{I}^+[t] \cup \mathsf{q}_t \mid \mathsf{q}_t \in \mathcal{Q}[t]\}$. By Prop. 6.1, each $\mathcal{I}^+[t] \cup \mathsf{q}_t$ is a valid $POMIS^+$ set at time $t$. Thus, every sequence in $\mathbb{S}$ consists of only $POMIS^+$ sets.

**(Completeness)** The algorithm recursively traces backward from $t = T$ to earlier time steps, guided by the smallest time index in the current IB$^+$ map, denoted $t_e$, and recurses with horizon $t_e - 1$. At each step, the algorithm considers all valid MUCT/IB pairs for $Y_t$. The correctness of this step is guaranteed by the soundness and completeness of the algorithm POMISs from Lee and Bareinboim [2018, Theorem 9]. Each recursive call updates the maps $\mathcal{I}^+$ and $\mathcal{Q}$ by accumulating IB$^+$ and QIB sets across all considered time steps. Once the earliest time index in IB$^+$ reaches $t = 0$, the recursion terminates. At this point, the algorithm forms the Cartesian product $\prod_{t \in \texttt{keys}(\mathcal{I}^+) \cap \texttt{keys}(\mathcal{Q})} \{\mathcal{I}^+[t] \cup \mathsf{q}_t \mid \mathsf{q}_t \in \mathcal{Q}[t]\}$, enumerating every possible combination across time. Since every possible IB$^+$ and QIB configuration is explored and retained, the final set $\mathbb{S}$ contains all valid sequences composed solely of POMIS$^+$ sets.

$\square$

# H    Algorithmic Characterization of POMIS$^+$

---

**Algorithm 3** Update $\mathcal{I}^+$ and $\mathcal{Q}$ from $\mathcal{I}^+$

---

1: **function** IB$^+$($\mathbb{V}, \mathbf{X}, \mathcal{I}^+$)
2: **for** each $X \in \mathbf{X}$ **do**
3:     Identify the time step $t$ such that $X \in \mathbf{V}_t$
4:     **if** $t \notin \mathcal{I}^+$ **then**
5:         $\mathcal{I}^+[t] \leftarrow [\,]$
6:     append $X$ to $\mathcal{I}^+[t]$
7: **return** $\mathcal{I}^+$

8: **function** QIB($\mathcal{G}, \mathbb{V}, \mathbf{T}, \mathcal{Q}, \mathcal{I}^+$)
9: **for** each $t \in \texttt{keys}(\mathcal{I}^+)$ **do**
10:     **if** $t \notin \mathcal{Q}$ **then**
11:         $G_t, \mathcal{Q}[t] \leftarrow \mathcal{G}[\mathbf{V}_t]_{\overline{\mathcal{I}^+[t]}}, [\,]$
12:         **for** each $\mathbf{X} \in$ POMISs($G_t, Y_t$) **do**
13:             **if** $\mathbf{X} \cap \mathbf{T} = \emptyset$ **then**
14:                 append $\mathbf{X}$ to $\mathcal{Q}[t]$
15: **return** $\mathcal{Q}$

---

In this appendix, we provide a detailed explanation of the algorithmic components used in the enumeration of POMIS$^+$ intervention sequences, as introduced in §7. At first, we define and clarify the role of the two internal maps, $\mathcal{I}^+$ and $\mathcal{Q}$, which correspond to the Interventional border for the subsequent time steps (IB$^+$) and qualified interventional borders (QIB), respectively.

**Map of interventional border for the subsequent time steps ($\mathcal{I}^+$)**    Given a POMIS candidate $\mathbf{X}$ obtained from the MUCT-IB procedure, we identify the time step $t$ associated with each variable $X \in \mathbf{X}$, and store it in the map $\mathcal{I}^+[t]$. The map $\mathcal{I}^+$ thus organizes variables in $\mathbf{X}$ by their corresponding time step, preserving the temporal alignment of possible interventions. This time-indexed representation allows the algorithm to recursively evaluate which time steps still require backward expansion, and is central to the structure of Alg. 1. This structure ensures that only the earliest relevant time step is explored recursively, avoiding redundant expansion into irrelevant subgraphs.

**Map of qualified interventional Border ($\mathcal{Q}$)**    Once $\mathcal{I}^+$ is populated, the QIB map $\mathcal{Q}$ is computed by evaluating each subgraph $\mathcal{G}[\mathbf{V}_t]$ after mutilating it with the intervention set $\mathcal{I}^+[t]$. From this mutilated subgraph, we re-run the POMISs algorithm on the local reward variable $Y_t$ to identify any remaining minimal intervention sets. These are stored in $\mathcal{Q}[t]$ only if they do not overlap with the current MUCT set $\mathbf{T}$, ensuring independence across temporal intervention levels. The QIB thus captures any residual variables at time $t$ that are necessary to optimize the local reward, complementing the already selected IB$^+$.

The recursive design of Alg. 1 ensures that the enumeration of intervention sequences terminates when all relevant intervention variables are assigned by time $t = 0$. At that point, the algorithm takes the Cartesian product of all IB$^+$ and QIB sets across time steps to generate complete intervention

sequences. This backward recursive approach avoids exhaustive enumeration by pruning irrelevant branches early and leveraging graph locality in the causal diagram.

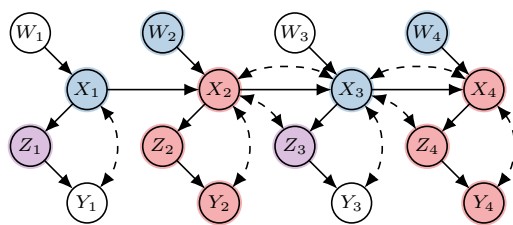

Figure 9: Illustration of the intermediate stage of the algorithm after two recursive calls.

**Example traces of recursive enumeration.** Fig. 9 illustrates an intermediate stage in the execution of our recursive POMIS$^+$ enumeration algorithm. The color-coded graph shows how MUCT, IB$^+$, and QIB evolve across time steps as the algorithm proceeds backward.

At the final time step $t = 3$ (corresponding to $Y_4$ in the figure), the algorithm computes the MUCT and IB from the subgraph $\mathcal{G}[\mathbf{V}_3 \cup \mathbf{V}_4]$, resulting in a MUCT set $\{X_4, Z_4, Y_4\}$ (highlighted in red). The corresponding IB$^+$ consists of the set $\{X_3, W_4\}$, indicated in blue. Next, the QIB is computed by evaluating the mutilated subgraph $\mathcal{G}[\mathbf{V}_3]_{\overline{X_3, W_4}}$, yielding $\{Z_3\}$ as an intervention set (highlighted in purple).

Since the current earliest time step in IB$^+$ is $t = 3 > 1$, the algorithm proceeds recursively with $T \leftarrow 2$. In this second round, the algorithm again computes the MUCT/IB pair for $Y_2$, resulting in MUCT $\{X_2, Z_2, Y_2\}$ (red), IB$^+ = \{X_1, W_2\}$ (blue), and QIB = $\{Z_1\}$ (purple), after mutilating $\mathcal{G}[\mathbf{V}_1]$ with $\{X_1, W_2\}$.

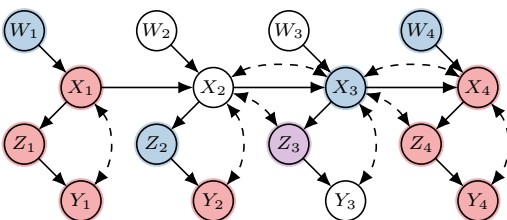

Figure 10: Illustration of the intermediate stage of the algorithm after three recursive calls.

Fig. 10 also provides one of the recursive traces of our POMIS$^+$ enumeration algorithm over steps, starting from the final reward $Y_4$ at time $t = 4$ and proceeding backward to $Y_1$ at $t = 1$. As before, red nodes indicate variables in the MUCT set, blue nodes indicate IB$^+$ variables and purple nodes denote QIB elements at each stage.

At time $t = 4$, the algorithm identifies the MUCT $\{Y_4, Z_4, X_4\}$ (highlighted in red), computed from the subgraph $\mathcal{G}[\mathbf{V}_3 \cup \mathbf{V}_4]$. The IB$^+$ map is then updated to include $\{X_3, W_4\}$ (blue), and since $\{Z_3\}$ influences to $Y_3$ in $\mathcal{G}[\mathbf{V}_3]_{\overline{X_3, W_4}}$, the QIB is $\{Z_3\}$ at this step.

Proceeding to time $t = 2$, the algorithm observes that a single node $Y_3$ can be a MUCT. The corresponding IB$^+$ is updated to $\{Z_2\}$ (purple), with no further QIB identified. At time $t = 1$, the MUCT set becomes $\{Y_1, Z_1, X_1\}$ (red). The IB$^+$ is now $\{Z_1\}$ (blue), as $Z_1$ is needed to activate paths to $Y_2$ under the current mutilated graph. Again, no QIB is generated at this time. Finally, at $t = 0$, the algorithm computes MUCT as $\{X_1, Z_1, Y_1\}$ (red), and identifies IB$^+ = \{W_1\}$ (blue), completing the backward exploration.

This example highlights how our algorithm incrementally expands the maps $\mathcal{I}^+$ and $\mathcal{Q}$ over time by tracing from $Y_T$ back to earlier rewards. Each recursive call explores a new MUCT/IB combination while updating time-specific intervention targets, eventually enabling full enumeration when the earliest IB$^+$ reaches time $t = 1$.

**Time Complexity of Alg. 1.** The recursive construction of POMIS$^+$ sequences explores possible intervention combinations over a time horizon $T$. In the worst case, each time step requires enumer-

ating subsets of ancestors of $Y_t$ to compute MUCT/IB pairs, followed by QIB exploration—both involving searches over subsets of the variable set $\mathbf{V}$ (i.e., $\mathcal{O}(2^{2|\mathbf{V}|})$). Each of these steps incurs an exponential cost of up to $2^{|\mathbf{V}|}$, resulting in $2^{2|\mathbf{V}|}$ complexity per time step. After $T$ recursion, the total worst-case time complexity becomes $\mathcal{O}(2^{2|\mathbf{V}| \cdot T})$.

# I Experiment Details

In this section, we provide detailed specifications of the SCMs used in our experiments to ensure reproducibility. For each experiment, the simulation was repeated 200 times using the corresponding SCM. All experiments were run on a dual-socket Intel Xeon Gold 5317 system with 24 physical cores (48 logical threads) at 3.0GHz.

**Task 1.** We implement an SCM composed of three time steps $t = 1, 2, 3$ with variable sets $\{X_t, Z_t, Y_t\}$ and structural equations defined to model the propagation of intervention effects over time. The probability distributions over exogenous variables $\mathbf{U}$ are defined as: We use the following probabilities over exogenous variables:

$$P(U_{X_1} = 1) = 0.85, \quad P(U_{X_1 Y_1} = 1) = 0.47, \quad P(U_{Z_1} = 1) = 0.14, \quad P(U_{Y_1} = 1) = 0.02,$$
$$P(U_{X_2} = 1) = 0.80, \quad P(U_{X_2 Y_2} = 1) = 0.55, \quad P(U_{Z_2} = 1) = 0.14, \quad P(U_{Y_2} = 1) = 0.01,$$
$$P(U_{X_3} = 1) = 0.01, \quad P(U_{X_3 Y_3} = 1) = 0.51, \quad P(U_{Z_3} = 1) = 0.13, \quad P(U_{Y_3} = 1) = 0.05.$$

The structural functions $f_V$ for each endogenous variable $V$ are as follows (where $\oplus$ denotes binary XOR and $v$ is the valuation of its parents):

$$f_{X_1} = U_{X_1} \oplus U_{X_1 Y_1} \qquad f_{X_2} = X_1 \oplus U_{X_2} \oplus U_{X_2 Y_2} \qquad f_{X_3} = X_2 \oplus U_{X_3} \oplus U_{X_3 Y_3}$$
$$f_{Z_1} = X_1 \oplus U_{Z_1} \qquad f_{Z_2} = X_2 \oplus U_{Z_2} \qquad f_{Z_3} = X_3 \oplus U_{Z_3}$$
$$f_{Y_1} = Z_1 \oplus U_{Y_1} \oplus U_{X_1 Y_1} \qquad f_{Y_2} = Z_2 \oplus U_{Y_2} \oplus U_{X_2 Y_2} \qquad f_{Y_3} = Z_3 \oplus U_{Y_3} \oplus U_{X_3 Y_3}$$

$$(a)\ t = 1 \qquad\qquad\qquad (b)\ t = 2 \qquad\qquad\qquad (c)\ t = 3$$

Figure 11: SCM definition for Task 1.

**Task 2.** We conducted the experiment with a two-step SCM defined over variables $\{W_t, X_t, Z_t, Y_t\}$ for $t = 1, 2$, as illustrated in Fig. 4. The structure explicitly models how intervention effects on $X_t$ propagate across time via $X_{t+1}$ and influence downstream outcomes $Y_t$ and $Y_{t+1}$ through intermediary variables $Z_t$.

The exogenous distribution $P(\mathbf{U})$ is parameterized to highlight the long-term influence of early interventions. The assigned probabilities are as follows:

$$P(U_{X_1} = 1) = 0.15, \quad P(U_{Z_1} = 1) = 0.15, \quad P(U_{Y_1} = 1) = 0.02, \quad P(U_{X_1 Y_1} = 1) = 0.47$$
$$P(U_{W_1} = 1) = 0.15, \quad P(U_{X_2} = 1) = 0.02, \quad P(U_{Z_2} = 1) = 0.12, \quad P(U_{X_2 Y_2} = 1) = 0.55$$
$$P(U_{Y_2} = 1) = 0.01, \quad P(U_{W_2} = 1) = 0.12.$$

We define the structural equations in Fig. 12 (where $\oplus$ denotes binary XOR). Notably, the structural

$$f_{W_0} = U_{W_0} \qquad\qquad\qquad f_{W_1} = U_{W_1}$$
$$f_{X_0} = U_{X_0} \oplus W_0 \oplus U_{X_0 Y_0} \qquad\qquad f_{X_1} = U_{X_1} \oplus W_1 \oplus U_{X_1 Y_1} \oplus X_0$$
$$f_{Z_0} = X_0 \oplus U_{Z_0} \qquad\qquad\qquad f_{Z_1} = X_1 \oplus U_{Z_1}$$
$$f_{Y_0} = Z_0 \oplus U_{Y_0} \oplus U_{X_0 Y_0} \qquad\qquad f_{Y_1} = Z_1 \oplus U_{Y_1} \oplus U_{X_1 Y_1}$$

$$(a)\ t = 0 \qquad\qquad\qquad\qquad\qquad (b)\ t = 1$$

Figure 12: SCM definition for Task 2.

functions for $X_2$ and $Y_2$ include $X_1$ as a parent, creating a direct dependency between early decisions and future rewards. This design allows us to test the effectiveness of DUC-based POMIS$^+$ strategies in capturing delayed causal influences that are missed by myopic approaches.

**Task 3.** To clearly illustrate how information from early IB$^+$ and QIB components can propagate across multiple time steps in the absence of the time-slice Markov assumption (Assumption 3.1), we design an SCM inducing Fig. 6. This structure focuses on exposing the long-range effect of early interventions. The SCM contains three time steps $t = 1, 2, 3$ over variables $\{W_t, X_t, Y_t\}$, with unobserved confounding between $X_t$ and $Y_t$ at each step. A bidirected edge between $X_2$ and $X_3$ further emphasizes the departure from the time-slice Markov assumption. This configuration enables us to test whether POMIS$^+$ can successfully capture reward-relevant information originating from earlier steps.

The structural functions $f_V$ for each endogenous variable $V$ are defined as follows:

$$f_{W_1} = U_{W_1}$$
$$f_{X_1} = U_{X_1} \oplus W_1 \oplus U_{X_1 Y_1}$$
$$f_{Y_1} = X_0 \oplus U_{Y_1} \oplus U_{X_1 Y_1}$$

(a) $t = 1$

$$f_{W_2} = U_{W_2}$$
$$f_{X_2} = U_{X_2} \oplus W_2 \oplus U_{X_2 X_3}$$
$$\oplus U_{X_2 Y_2} \oplus X_1$$
$$f_{Y_2} = X_2 \oplus U_{Y_2} \oplus U_{X_2 Y_2}$$

(b) $t = 2$

$$f_{W_3} = U_{W_3}$$
$$f_{X_3} = U_{X_3} \oplus W_3 \oplus U_{X_2 X_3}$$
$$\oplus U_{X_3 Y_3} \oplus X_2$$
$$f_{Y_3} = X_3 \oplus U_{Y_3} \oplus U_{X_3 Y_3}$$

(c) $t = 3$

Figure 13: SCM definition for Task 3.

The probability distributions over exogenous variables $\mathbf{U}$ are defined as:

$P(U_{X_1} = 1) = 0.82,$ $\quad P(U_{W_1} = 1) = 0.15,$ $\quad P(U_{X_1 Y_1} = 1) = 0.52,$ $\quad P(U_{X_2} = 1) = 0.92,$
$P(U_{Y_2} = 1) = 0.02,$ $\quad P(U_{X_2 Y_2} = 1) = 0.44,$ $\quad P(U_{X_2 X_3} = 1) = 0.43,$ $\quad P(U_{Y_2} = 1) = 0.01,$
$P(U_{W_2} = 1) = 0.42,$ $\quad P(U_{X_3} = 1) = 0.20,$ $\quad P(U_{X_3 Y_3} = 1) = 0.48,$ $\quad P(U_{Y_3} = 1) = 0.05,$
$P(U_{W_3} = 1) = 0.41.$

# J    Limited Alternative Methods to Approach on NS-SCM-MAB

## J.1    Projection-based approach on NS-SCM-MAB

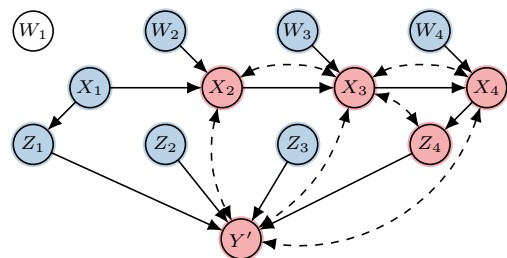

Figure 14: MUCT and IB in the $\mathcal{G}_{[\mathbf{V} \setminus \mathbf{Y}]}$

Lee and Bareinboim [2019] builds on the framework of Lee and Bareinboim [2018] by relaxing the assumption that every variable in the system must be manipulable, while still operating under a stationary setting. Their method uses latent projections (Verma and Pearl [1990], Verma [1992]) to identify possibly-optimal intervention targets and infer dependency relations among them, even when only partial causal information is available.

In our setting, although the underlying process is non-stationary, the complete causal graph is revealed after all rounds. This allows the latent projection technique to be applied post hoc for identifying possibly-optimal arms. Nevertheless, such an approach overlooks the temporal dynamics inherent to the non-stationary process, thereby failing to capture how specific interventions affect rewards across different time steps. In what follows, we compare this projection-based approach with our method.

First of all, we introduce the projection-based approach. In a stationary setting with non-manipulable variables, Lee and Bareinboim [2019] characterize possibly-optimal arms using the latent projection technique [Verma and Pearl, 1990, Verma, 1992]. They begin with a causal diagram $\mathcal{G} = \langle \mathbf{V}, \mathbf{E} \rangle$, and define a set of non-manipulative variables $\mathbf{N} \subset \mathbf{V} \setminus \{Y\}$, where $Y$ is the target variable. To construct

the latent projection onto the manipulative variables $\mathbf{V} \setminus \mathbf{N}$, they consider an augmented graph $\widehat{\mathcal{G}}$ that explicitly represents unobserved confounders. They initialize a graph $\mathcal{H} = \langle \mathbf{V} \setminus \mathbf{N}, \emptyset \rangle$, then add edges as follows:

1. A directed edge between $V_i$ and $V_j$ if $V_i \rightarrow V_j \in \mathcal{G}$ or there exists a directed path from $V_i$ to $V_j$ where all non-end vertices in the path between there are in $\mathbf{N}$.

2. A bi-directed edge between $V_i$ and $V_j$ if $V_i \leftrightarrow V_j \in \mathcal{G}$; or there exists directed paths from an unobserved confounder to $V_i$ and $V_j$ in $\hat{\mathcal{G}}$ where all non-end vertices are in $\mathbf{N}$.

Let $\mathcal{G}_{[\mathbf{V}']}$ denote the causal diagram resulting from projecting $\mathcal{G}$ onto $\mathbf{V}'$. They prove that $\mathbb{P}^{\mathbf{N}}_{\mathcal{G},Y} = \mathbb{P}_{\mathcal{H},Y}$ (i.e., POMISs given $\langle \mathcal{G}, Y, \mathbf{N} \rangle$ = POMISs given $\langle \mathcal{H}, Y \rangle$) via two propositions, which ensures that the optimality of an arm remains under (i) projection from $\mathcal{G}$ to $\mathcal{H}$ and (ii) the reverse projection.

**Proposition J.1** (Causal Identification without Non-manipulative Variables [Lee and Bareinboim, 2019]). *Given an SCM $\mathcal{M}^1 = \mathcal{M} = \langle \mathbf{V}, \mathbf{U}, \mathcal{F}, P(\mathbf{U}) \rangle$, there exists an SCM $\mathcal{M}^2 = \mathcal{M}_{\mathbf{V} \setminus \mathbf{N}} = \langle \mathbf{V}, \mathbf{U}, \mathcal{F}', P(\mathbf{U}) \rangle$ such that $P^1_{\mathbf{x}}(\mathbf{y}) = P^2_{\mathbf{x}}(\mathbf{y})$ for any $\mathbf{X}, \mathbf{Y} \subset \mathbf{V} \setminus \mathbf{N}$ and $\mathbf{Y} \neq \emptyset$.*

**Proposition J.2** (Causal Identification under the Projected Graph [Lee and Bareinboim, 2019]). *Given a causal diagram $\mathcal{G}$, let $\mathcal{H} = \mathcal{G}_{[\mathbf{V} \setminus \mathbf{N}]}$. For a SCM $\mathcal{M}^1 = \mathcal{M}_{[\mathbf{V} \setminus \mathbf{N}]} = \langle \mathbf{V} \setminus \mathbf{N}, \mathbf{U}, \mathbf{F}', P(\mathbf{U}) \rangle$ conforming to $\mathcal{H}$, there exists a SCM $\mathcal{M}^2 = \mathcal{M} = \langle \mathbf{V}, \mathbf{U}, \mathbf{F}, P(\mathbf{U}) \rangle$ that conforms to $\mathcal{G}$ such that $P^1_{\mathbf{x}}(\mathbf{y}) = P^2_{\mathbf{x}}(\mathbf{y})$, for any $\mathbf{X}, \mathbf{Y} \subseteq \mathbf{V} \setminus \mathbf{N}$ and $\mathbf{Y} \neq \emptyset$.*

**Theorem J.1** (POMIS Invariance under Projection). *Given a causal diagram $\mathcal{G} = \langle \mathbf{V}, \mathbf{E} \rangle$, a reward variable $Y \in \mathbf{V}$, and a set of non-manipulable variables $\mathbf{N} \subseteq \mathbf{V} \setminus \{Y\}$, let $\mathcal{H}$ be the projection of $\mathcal{G}$ onto $\mathbf{V} \setminus \mathbf{N}$. Then,*

$$\mathbb{P}^{\mathbf{N}}_{\mathcal{G},Y} = \mathbb{P}_{\mathcal{H},Y}$$

*where $\mathbb{P}^{\mathbf{N}}_{\mathcal{G},Y}$ denotes a set of POMISs given $\langle \mathcal{G}, Y, \mathbf{N} \rangle$.*

Based on the projection-based approach introduced above, we can derive Fig. 8 via the latent projection. In Fig. 14, we obtain POMIS $= \{Z_1, X_1, Z_2, W_2, Z_3, W_3, W_4\}$. This result coincides with the outcome of our proposed POMIS$^+$-based method (Fig. 8) in terms of maximizing the cumulative reward $Y' = \sum_{t=1}^{4} Y_t$. However, since the graph in Fig. 14 is constructed by projecting all temporal models into a single static, stationary structure, it lacks temporal interpretability. We argue that such a projection is insufficient for modeling and analyzing non-stationary bandit problems, where the notion of time is inseparable from the causal dynamics.

Non-stationarity in bandits inherently involves modeling how reward distributions evolve over time, and our method is specifically designed to capture and exploit this temporal evolution. The significant advantage of our framework lies in its ability to identify time-specific intervention sets for each reward variable $Y_t$. This makes POMIS$^+$ not only optimal in terms of cumulative reward but also more interpretable and practically applicable to real-world scenarios where intervention constraints or objectives vary over time. Below, we present one example that highlights this property.

**Illustrative case: Sequential treatment with risk constraint** Suppose each time-indexed intervention variable $Z_t, X_t, W_t$ represents a distinct medication administered at time $t$:

- $Z_t$: fast-acting drug that strongly affects immediate health ($Y_t$) but may induce side effects.
- $X_t$: slow-acting drug that influences future outcomes (e.g., $Y_{t+1}, Y_{t+2}$).
- $W_t$: a supportive drug that amplifies or stabilizes drug effects in the future.

Now consider a scenario where we impose a safety constraint on early reward:

Let $\mathbf{A} = (\{Z_1, X_1\}, \{Z_2, W_2\}, \{Z_3, W_3\}, \{W_4\})$ denote the intervention sequence obtained from the POMIS$^+$ method (or projection-based method) across all time steps, and let $\mathbf{a}_0 \in \mathscr{D}(\mathbf{A}[0])$ be a joint intervention assignment to the first intervention set.

$$\mathbb{E}[Y_1 \mid \mathrm{do}(\mathbf{a}_0)] \leq \epsilon,$$

to ensure that early treatments do not induce excessive physiological stress. A projection-based method (Fig. 14) optimizes $\mathbb{E}[Y']$ and may select ($Z_1 = 1$) if it increases $Y'$ overall—despite the fact that $Z_1$ directly affects $Y_1$ and may violate the safety constraint.

In contrast, our method (Fig. 8) distinguishes that:

- $Z_1$ is $\text{QIB}_1$ influencing primarily $Y_1$, while
- $X_1$ is $\text{IB}_{1,4}^+$ that contributes to $Y_4$ through the path $X_1 \to X_2 \to X_3 \to \cdots \to Y_4$.

By leveraging this temporal decomposition, we can selectively intervene on $X_1$ while avoiding $Z_1$, satisfying the constraint on $Y_1$ and still improving long-term outcomes. Formally, we solve:

$$\mathbf{a}^* = \underset{\substack{\mathbf{a} \in \mathscr{D}(\mathbf{A}) \\ \text{s.t.} \mathbb{E}[Y_1|\text{do}(\mathbf{a}_0)] \le \epsilon}}{\arg\max} \quad \mathbb{E}\left[ \sum_{t=1}^{4} Y_t \mid \text{do}(\mathbf{a}) \right]$$

**Summary**   This example highlights the limitations of projection in non-stationary bandits. While projection-based methods may yield high cumulative rewards, they are blind to when and how specific interventions act. Our $\text{POMIS}^+$ framework enables temporally structured intervention planning, allowing for interpretable, constraint-aware, and sequentially optimal policies. In domains such as medicine or education—where interventions at each stage must consider safety or ethical constraints—such temporal disentanglement is essential.

## J.2   Forced stationary approach on NS-SCM-MAB



(a) $\mathcal{G}$        (b) $\mathcal{G}_{\overline{X_1, X_2}}$        (c) $\mathcal{G}_{\overline{X_1, W_2}}$

Figure 15: (a) Non-stationary causal diagram $\mathcal{G}$ and (b, c) mutilated graphs

In the main text, we analyzed the non-stationary SCM-MAB setting in which the reward distribution changes over time due to the causal influence of past interventions (see Def. 3.1). While our main algorithmic framework leverages time-specific causal structure, it is also possible to consider an alternative approach: *forced stationarity* via direct intervention. The idea is to identify intervention sets that block the information propagation (see §3) across time slices—thus preventing reward distribution shifts. This allows the agent to reuse previously learned interventional effects, reducing the need to re-identify POMIS sets or re-calculate expected rewards at every time step. Before formalizing this concept, we introduce the notion of transition edge blocking.

**Definition J.1** (Block). Let $\mathcal{G}$ be a causal diagram and $\mathbf{V}_t$ the set of endogenous variables at time $t$. An intervention $\text{do}(\mathbf{X}_t = \mathbf{x}_t)$ is said to block transition edges if, in the mutilated graph $\mathcal{G}_{\overline{\mathbf{X}_t}}$, all incoming edges from variables in previous time slices $\mathbf{V}_{<t}$ to $\mathbf{X}_t$ are removed. This operation prevents the information propagation from $\mathbf{V}_{<t}$ to $\mathbf{V}_t$ via $\mathbf{X}_t$.

When such a blocking intervention is applied, the reward variable $Y_t$ becomes conditionally isolated from its historical influences. This motivates the following definition of forced stationarity, grounded in the interventional reward distributions formalized in Def. 3.1.

**Definition J.2** (Forced Stationarity). Given a non-stationary SCM-MAB $\langle \mathcal{M}, \mathbf{Y} \rangle$, we say that an intervention set $\mathbf{F} \subseteq \mathbf{V}_t$ induces forced stationarity over reward variable $Y_t$ if for every $t < t'$, and for every $\mathbf{f} \in \mathscr{D}(\mathbf{F})$,

$$P\left(Y_t \mid \text{do}(\mathbf{f}), \mathbb{1}_{t>1} \cdot I_{1:t-1}\right) = P\left(Y_{t'} \mid \text{do}(\mathbf{f}), \mathbb{1}_{t'>1} \cdot (I_{1:t-1} \cup I_{t:t'-1})\right),$$

In other words, the reward distribution remains invariant over time under repeated interventions on $\mathbf{F}$.

This formalization allows us to distinguish between *actual* stationarity (inherent in the SCM) and stationarity that is induced by intervention. The following lemma clarifies the relationship between blocking transition edges and enforcing reward stationarity.

**Lemma J.1** (Blocking transition edges induces forced stationarity). *Let $\mathcal{G}$ be a causal diagram, and suppose an intervention on $\{Z_t\}$ blocks the transition edge $Z_{t-1} \to Z_t$. Then, we have:*

$$P\left(Y_{t-1} \mid \mathrm{do}(Z_{t-1}), \mathbb{1}_{t>1} \cdot I_{1:t-2}\right) = P\left(Y_t \mid \mathrm{do}(Z_t), \mathbb{1}_{t>1} \cdot I_{1:t-1}\right).$$

*Therefore, the intervention enforces reward stationarity across time.*

*Proof.* By intervening on $Z_t$, we remove the transition edge from $Z_{t-1}$ to $Z_t$ (Def. J.1), thereby preventing any causal effect from prior interventions from propagating to $Z_t$. As a result, $Y_t$ under $\mathrm{do}(Z_t)$ becomes conditionally independent of earlier actions, and its interventional distribution same as that of $Y_{t-1}$ under $\mathrm{do}(Z_{t-1})$. □

The idea of forced stationarity provides a mechanism to simplify learning in non-stationary SCM-MABs: if one can construct a policy that always intervenes on such blocking variables (e.g., $Z_t$), the resulting reward distribution no longer shifts over time. This allows the agent to reuse previously learned information without recalculating POMISs at each step.

However, such interventions may reduce the optimality of the resulting policy. Although reward distributions may appear stationary under intervention $\mathbf{F}$, the values themselves may not be maximized. Blocking information from previous time slices may prevent the agent from exploiting long-term causal pathways that yield higher rewards.

To illustrate this trade-off, consider the causal graphs in Fig. 15. In Fig. 15(b), we apply an intervention on $X_2$, which removes the incoming edge from $X_1$ to $X_2$, thereby blocking temporal information transfer and enforcing stationarity. While this simplifies the learning problem, it overlooks alternative intervention options. In Fig. 15(c), we do not intervene on $X_2$, but instead on $W_2$—a variable that is structurally valid and potentially in maximizing reward, even though it does not block the transition path.

This example reveals an important limitation of the forced stationarity paradigm: sets such as $\{X_2\}$ and $\{W_2\}$ both are interventional border (IB), yet only $X_2$ induces stationarity when intervened. If one naively prioritizes only those IB variables that block transition edges, the policy may become overly myopic and ignore possibly higher-rewarding options. In contrast, our framework (i.e., POMIS$^+$) retains the full temporal structure and dynamically adapts to changes in reward distribution without forcibly blocking information. This enables both fine-grained interpretability and optimal intervention selection over time.

While the limitations of forced stationarity are evident in scenarios where blocking discards useful long-term causal dependencies, we emphasize that this strategy is not inherently flawed. In fact, under certain structural conditions, forced stationarity may serve as a practical approach to simplifying learning in temporally complex environments. For example, if the reward variable $Y_t$ consistently depends on a repeated set of parent variables over time—such as through recurring transition edges—then blocking those transitions can yield invariant reward distributions without sacrificing performance.

In such cases, domain knowledge or structural priors can inform targeted interventions that induce stationarity while still capturing the most influential causal paths. This highlights a broader point: rather than rejecting forced stationarity entirely, it can be selectively applied when supported by sufficient knowledge about the system's temporal regularities.

# K   More Discussions

## K.1   Relaxing Natural Characteristics on Temporal Structure

Our main theoretical development naturally assumes both an identical graphical structure across time slices and the existence of transition edges between them. These assumptions are introduced to reflect natural characteristics of non-stationary environments and to facilitate tractable inference. Importantly, the assumption of identical structure is not essential to the correctness of our algorithm: even when the assumption is relaxed, our method remains sound and complete, provided that the full temporal causal graph is accessible.

In contrast, the existence of transition edges plays a more critical role. Without such edges, causal information cannot propagate across time, and the algorithm may fail to identify relevant variables at earlier steps, limiting its ability to construct non-myopic intervention strategies.

- The property of identical time slices reflects the natural structure of stationary bandit models (see, Fig. 1(b)), where variables and their dependencies are replicated across time for each time slice. While this natural assumption simplifies the search space by enabling repeated reasoning patterns, which is a characteristic of bandit settings, it is not required for the correctness of the graphical analysis or the definitions of $IB^+$ and QIB.
- The existence of transition edges between time slices enables information to propagate forward in time, allowing the algorithm to identify earlier variables that support future interventions. Without transition edges, information cannot propagate across time, preventing the algorithm from identifying long-range dependencies or constructing effective non-myopic strategies. More concretely, certain paths to earlier supporting variables (i.e., $IB^+$) become unreachable, limiting the algorithm's ability to exploit temporal structure for long-term optimization.

This flexibility ensures that our framework is broadly applicable, even in domains where temporal structure varies over time. We emphasize that these assumptions serve as modeling conveniences that facilitate analysis and interpretation, rather than strict algorithmic requirements.

## K.2 Non-Stationarity in Causal Bandits vs. Causality in Non-Stationary Bandits

It is useful to distinguish our formulation from alternative approaches that combine causality and non-stationarity in opposite directions. Our work builds upon structural causal bandits (SCM-MABs) [Lee and Bareinboim, 2018, 2019], which assume a fixed underlying causal graph and utilize it to guide interventions. We extend this line of work by introducing non-stationarity at the level of the causal structure itself—modeling how causal dependencies and reward mechanisms evolve over time. In this sense, we develop a causal bandit framework that internalizes non-stationarity as a structural property.

By contrast, recent efforts that introduce causal reasoning into non-stationary bandit settings typically begin with traditional statistical formulations—such as sliding-window or change-point models—and incorporate causal estimators to correct for confounding or adapt to changing environments (details are available in Appendix E). For instance, Huang and Wu [2024] propose a method for handling confounded and selection-biased offline data by deriving robust causal bounds for each arm. Similarly, Nourani-Koliji et al. [2023] address piecewise-stationary bandits with causally related rewards by detecting changes in both reward distributions and causal structure using a Generalized Likelihood Ratio(GLR)-based change-point detector. In these settings, causality is layered onto a fundamentally statistical treatment of temporal change, often without structural assumptions about the underlying dynamics.

The distinction is more than philosophical. Our model-based approach enables:

- Reasoning about how non-stationarity arises (via temporal models and transition edges),
- Derivation of non-myopic intervention strategies through temporally-aware graphical structures ($POMIS^+$), and
- Theoretical guarantees grounded in structural assumptions.

In contrast, approaches based on statistical modeling typically emphasize adaptation—e.g., through sliding windows, discounting, or change-point detection—without modeling the underlying structural mechanisms that generate non-stationarity. These methods act as black-box in the sense that they detect shifts or trends in the data but do not explain them in terms of causal relationships or system dynamics.

## K.3 $POMIS^+$ Sequences as Composition of Possibly Optimal Intervention Sets within Temporal Models

An intervention sequence and the causal responses of each time-indexed variable determine the expected cumulative reward in a sequential decision problem. In our framework, this is formal-

ized by an SCM $\mathcal{M}$ together with a sequence of possibly-optimal intervention sets—namely, POMIS$^+$—identified across time.

While SCM-level definitions of POMIS (cf. Def. B.2) evaluate optimality based on the existence of at least one SCM where a given intervention set performs optimally, temporal decision-making imposes a stricter constraint: each reward $Y_t$ must be evaluated in the context of prior interventions. This context naturally gives rise to a family of temporal models $\mathcal{M}_t \mid \mathbf{v}_t^\star$, where $\mathbf{v}_t^\star$ represents the predetermined values induced by earlier actions. Thus, the optimal intervention at each time step may be determined within this temporal model, reflecting only the information propagation available at time $t$.

In this subsection, we show that the expected cumulative reward under a full SCM can be equivalently decomposed into a sum of expected rewards under temporal models. Each such term corresponds to an intervention on a POMIS$^+$ set at time $t$, selected with respect to $\mathcal{M}_t \mid \mathbf{v}_t^\star$. The result is formalized in Prop. K.1, which clarifies how the temporal roll-out of POMIS$^+$ constructs a compositionally valid and possibly-optimal global intervention plan.

**Proposition K.1** (Composition of Temporal Optima). *Let $\mathcal{M}$ be an SCM conforming to $\mathcal{G}$, and let $(\mathbf{X}_1, \ldots, \mathbf{X}_T)$ be an optimal intervention sequence with realizations $\mathbf{x}_1^*, \ldots, \mathbf{x}_T^*$. Let $\mathbf{v}_t^\star$ denote the predetermined variables at time $t$ given past interventions. Suppose each intervention set $\mathbf{X}_t$ is constructed as POMIS$_t \cup \mathbf{W}_t$, where $\mathbf{W}_t \subseteq Pa(\mathbf{V}_{t'})$ satisfies the condition in Thm. 5.1. Then:*

$$\mathbb{E}^{\mathcal{M}} \left[ \sum_{t=1}^{T} Y_t \mid \mathrm{do}(\mathbf{x}_1^*, \ldots, \mathbf{x}_T^*) \right] = \sum_{t=1}^{T} \mathbb{E}^{\mathcal{M}_t \mid \mathbf{v}_t^\star} \left[ Y_t \mid \mathrm{do}(\mathbf{x}_t^*) \right]$$

*Proof.* We begin by expanding the expected cumulative reward under the full SCM $\mathcal{M}$:

$$\mathbb{E}^{\mathcal{M}} \left[ \sum_{t=1}^{T} Y_t \mid \mathrm{do}(\mathbf{x}_1^*, \ldots, \mathbf{x}_T^*) \right] = \sum_{t=1}^{T} \mathbb{E}^{\mathcal{M}} \left[ Y_t \mid \mathrm{do}(\mathbf{x}_1^*, \ldots, \mathbf{x}_T^*) \right].$$

For each $t \in [T]$, let $\mathbf{v}_t^\star$ denote the predetermined assignment to $Pa(\mathbf{V}_t)$ induced by prior interventions $\mathrm{do}(\mathbf{x}_1^*, \ldots, \mathbf{x}_{t-1}^*)$. Then the reward at time $t$ can be equivalently computed in the conditioned temporal model $\mathcal{M}_t \mid \mathbf{v}_t^\star$ as:

$$\mathbb{E}^{\mathcal{M}} \left[ Y_t \mid \mathrm{do}(\mathbf{x}_1^*, \ldots, \mathbf{x}_T^*) \right] = \mathbb{E}^{\mathcal{M}_t \mid \mathbf{v}_t^\star} \left[ Y_t \mid \mathrm{do}(\mathbf{x}_t^*) \right].$$

Applying this to each $t$ and summing, we obtain:

$$\mathbb{E}^{\mathcal{M}} \left[ \sum_{t=1}^{T} Y_t \mid \mathrm{do}(\mathbf{x}_1^*, \ldots, \mathbf{x}_T^*) \right] = \sum_{t=1}^{T} \mathbb{E}^{\mathcal{M}_t \mid \mathbf{v}_t^\star} \left[ Y_t \mid \mathrm{do}(\mathbf{x}_t^*) \right],$$

as claimed. $\qquad\square$

This proposition formalizes how the cumulative reward in a full SCM can be decomposed into a sequence of rewards, each governed by a temporal model conditioned on prior interventions (i.e., predetermined values). It supports the view that the POMIS$^+$ sequence acts as a possibly-optimal plan across time, even though the optimality of individual interventions may depend on specific prior values.

While each temporal model $\mathcal{M}_t \mid \mathbf{v}_t^\star$ determines the optimal intervention at time $t$ under a fixed history of prior interventions, the globally optimal sequence across the entire SCM must be selected jointly, since each temporal model depends on the conditioning induced by earlier decisions.

**Corollary K.1** (Local Optimality Does Not Imply Global Optimality). *There exists an SCM $\mathcal{M}$ and intervention sequences $(\mathbf{X}_1, \ldots, \mathbf{X}_T)$ such that for each $t$, $\mathbf{X}_t$ is a possibly-optimal intervention set in the temporal model $\mathcal{M}_t \mid \mathbf{v}_t^\star$, but the joint sequence $(\mathbf{X}_1, \ldots, \mathbf{X}_T)$ is not globally optimal for $\mathbb{E}^{\mathcal{M}} \left[ \sum_{t=1}^{T} Y_t \right]$.*

This corollary illustrates that local optimality under temporal models does not guarantee global optimality in the full SCM. Due to the interdependencies between time steps, early interventions may have long-term consequences that are not captured by myopic optimization. Therefore, joint planning across time is necessary to achieve globally optimal decisions.

