# OpenReview forum: "Non-Stationary Structural Causal Bandits"
_NeurIPS.cc/2025/Conference — NeurIPS 2025 poster_

### Official Review · Reviewer_4V3R · 2025-06-06

**Clarity:** 2
**Significance:** 2
**Originality:** 3
**Rating:** 4
**Confidence:** 2

**Summary:**

This paper studies sequential decision making where rewards may be correlated through unobserved counfounders and across time. The relationship between variables are described through a temporal causal graph. The authors propose the POMIS+ approach, which allows to determine variables that contribute to maximizing immediate and future rewards, and test its performance in a numerical experiment.

**Questions:**

- What practical situations could be effectively modelled by your framework?

- Adversarial bandit theory is able to deal with arbitrarily complex reward structures. Although your non-stationary causal framework is certainly able to capture complex reward dynamics, it seems hard to obtain meaningful convergence results, as opposed to adversarial bandit theory for which several sharp theoretical results have been derived. What are the advantages of your method as compared to the adversarial bandit literature?

**Ethical Concerns:**

["NO or VERY MINOR ethics concerns only"]

**Final Justification:**

I am largely convinced by the additional clarifications they provided regarding the motivation and relevance of the proposed framework. In particular, I agree that applications involving sensitive policy-making require an understanding of complex causal structures, rather than simply adapting to observed payoff variations.

While I still find the proposed framework somewhat overly complex, and the results not particularly sharp, I recognize the potential value in laying the foundations for a new line of theoretical research. I appreciate the authors' effort to position their work as a step toward broader developments within the community.

In light of the authors’ commitment to better motivate their framework and improve the presentation—especially when introducing new concepts—I have decided to increase my score from 3 to 4.

**Limitations:**

I think this article could be of interest, however it is hardly legible in its current format because it is too complex and compact. I think that a conference like Neurips is not a good match for such a study: a specialized journal, where the authors would have more space to elaborate on their framework and its advantages would be better. I therefore lean towards rejecting it.

**Quality:**

3

**Strengths And Weaknesses:**

STRENGTHS:

- The framework is interesting is new.
- The authors did a great job putting the evolving causal environment in perspective with the existing literature in the supplementary material.

WEAKNESSES:

- The pratical interest of the framework is not sufficiently highlighted.

- I think the framework developed in this paper is too complicated and general for a conference paper. This comes with two inconvenients.

First, there are a lot of heavy notations which hamper the legibility of the paper: section 2 introduces too many concepts and notations without enough explanation in my opinion (manipulative vs non manipulative, which adds to endogenous vs exogenous variables...), and too many definitions (among which IB and QIB, which are quite intricate) are introduced after that. In my opinion, the 9 pages format is not suited for introducing such a complex machinery.

Second, the framework is too general for sharp theoretical insights to be derived, although assumption 3.1 and 3.2 significantly narrow it (which begs the question: why being so general before introducing such assumptions?). From my understanding, the results established in this paper are rather weak: they are limited to the existence of an optimal partial assignement (theorem 4.1), the fact that POMIS+ is Markovian (proposition 4.1)---which seems a natural consequence of assumption 3.1 indeed---, characterization of POMIS+ via QIB (proposition 5.1) which I don't find really insightful.

I must admit that in spite of having carefully readed the paper, I haven't understood well its main points (in spite of being rather knowledgeable in bandit theory and acquainted with causality). I think this is because it is too complicated and compact: a specialized journal should be a better match for this work.

---

> ### Author Rebuttal · Authors · 2025-07-31
>
> > The pratical interest of the framework is not sufficiently highlighted.
>
> Thank you for this valuable observation. We acknowledge that the practical value of our framework may not have been emphasized sufficiently in the initial submission and appreciate the opportunity to clarify.
>
> Our work is directly motivated by real-world decision-making scenarios where both **reward dynamics and underlying causal relationships evolve over time**—for example, in healthcare, where a sequence of treatments affects future patient outcomes through time-varying physiological mechanisms, or in education, where pedagogical interventions affect learning trajectories in temporally dependent ways.
>
> Classical bandit algorithms (stationary or non-stationary) often treat temporal variation as a statistical artifact, whereas structural causal bandits focus on the reward shift that occurs due to an evolving causal structure. This leads to **non-myopic strategies** that are critical in domains where interventions have delayed or cumulative effects.
>
> We will revise the manuscript to include concrete real-world motivating examples (e.g., time-evolving treatment planning in healthcare) and better highlight how our algorithm can guide decision-making in these settings.
>
> > there are a lot of heavy notations which hamper the legibility of the paper: ...
>
> `Section 2` **introduces several foundational concepts and notations** that may initially appear dense or abrupt. Our intention was to formally define the setting with precision, especially given the layered nature of the non-stationary SCM-MAB framework. In particular, we will add brief examples when defining manipulable and exogenous variables. We believe that improving the exposition in `Section 2` will enhance the overall accessibility and clarity of the paper.
>
> > The framework is too general for sharp theoretical insights to be derived, although assumption 3.1 and 3.2 significantly narrow it.
>
> > the results established in this paper are rather weak : they are limited to the existence ..
>
> Thank you for your concerns. We respectfully disagree with the characterization that our results are limited to the three components you listed. In fact, the **main contribution of our work lies not just in each result (e.g., Propositions, Theorems, and Definitions) taken individually, but also in how these components are carefully integrated to enable the systematic construction and enumeration of all valid intervention sequences over time.** No existing solution or algorithm can construct these intervention sequences.
>
> Theorem 4.1 (Existence), Proposition 4.1 (Temporal dependency of POMIS$^+$), and Proposition 5.1 (graphical characterization via QIB and IB$^+$) are **not standalone contributions**, but rather serve as the **theoretical backbone for Algorithm 1**, which is our central result. This algorithm is not a heuristic or sampling procedure—it is a **sound and complete method** that exhaustively enumerates **all possibly-optimal sequences** of interventions across multiple time steps, based purely on structural properties of the causal graph.
>
> To demonstrate the limitations of applying existing stationary algorithms [1] without our algorithm enumerating sequences, we present a concrete failure case in our motivating example (`Section 3.1`). As shown in **Figure 4(a)**, applying a method designed under stationary assumptions leads to the exclusion of a crucial variable, $X_1$, from the intervention set. As a result, the learned policy **fails to converge entirely**, even as the number of trials increases. This outcome clearly illustrates that **ignoring temporally extended causal dependencies can result in significant performance degradation**.
>
> Achieving this required more than technical derivations: we had to redefine the intervention problem under non-stationary causal dynamics (Definition 3.1), develop novel graphical tools (Definition 5.1 and 5.2, based on Theorem 4.1), and formalize their compositional structure across time (Algorithm 1 and Theorem 6.1). These results are deeply interdependent and were **designed as a unified framework** to support principled, structure-aware sequential decision-making in evolving environments.
>
> To the best of our knowledge, **no prior work** has offered a formal mechanism to construct all such valid intervention sequences in non-stationary causal settings. We will revise the manuscript to better emphasize how the results you mentioned contribute jointly to this broader and more impactful goal. Thank you again for prompting this clarification.
>
> [1] Lee, S., & Bareinboim, E. (2018). Structural causal bandits: Where to intervene?. Advances in neural information processing systems, 31.
>
> > too many definitions (among which IB and QIB, which are quite intricate) are introduced after that. In my opinion, the 9 pages format is not suited for introducing such a complex machinery.
>
> > A specialized journal should be a better match for this work.
>
> While we appreciate the suggestion, we believe our work is well aligned with the NeurIPS community. The problem of planning optimal intervention sequences under evolving causal structures sits at the intersection of causal inference, sequential decision-making, and graphical modeling—core topics of interest at NeurIPS.
>
> Recent NeurIPS papers, such as [2–4], reflect the community's engagement with structurally rich, causally informed decision frameworks. We also made an effort to ensure accessibility: rather than presenting definitions upfront, we begin with a motivating example in Section 3.1 that naturally introduces the core challenges and guides the reader toward the formal concepts (e.g., POMIS$^+$, QIB). We ensure that the final version will be made contributions even clearer.
>
> [2] Kumor, D., Zhang, J., & Bareinboim, E. (2021). Sequential causal imitation learning with unobserved confounders. *Advances in Neural Information Processing Systems*, *34*, 14669-14680.
>
> [3] Elahi, M. Q., Ghasemi, M., & Kocaoglu, M. (2024). Partial structure discovery is sufficient for no-regret learning in causal bandits. *Advances in Neural Information Processing Systems*, *37*, 109066-109100.
>
> [4] Zhang, J., & Bareinboim, E. (2022). Online reinforcement learning for mixed policy scopes. *Advances in Neural Information Processing Systems*, *35*, 3191-3202.
>
> ***- Questions:***
>
> > What practical situations could be effectively modelled by your framework?
>
> Our framework is designed for sequential decision-making problems where both the reward distribution and the underlying causal structure change over time, and where **long-term causal reasoning** is essential.
>
> One example is longitudinal healthcare, where early treatments can influence future health outcomes via time-delayed pathways. For instance, prescribing a medication today (e.g., $X_1$) may affect not only immediate outcomes ($Y_1$) but also downstream outcomes ($Y_2$, $Y_3$, ...) through intermediate variables—captured by temporal causal dependencies in our model. Our framework enables the identification of **non-myopic intervention sequences** that account for such evolving structures.
>
> Similarly, in education or public policy, early-stage interventions (e.g., tutoring, incentives) can have delayed and structurally mediated impacts. In these settings, modeling time-varying causal dependencies is critical to designing effective long-term strategies under limited budgets.
>
> We will clarify and emphasize these practical implications in the revision.
>
> > Adversarial bandit theory is able to deal with arbitrarily complex reward structures. ... What are the advantages of your method as compared to the adversarial bandit literature?
>
> Thank you for this question. We agree that adversarial bandit theory provides strong regret guarantees under general and worst-case reward sequences. However, our work is driven by a fundamentally different goal: rather than merely adapting to changing reward dynamics, we aim to understand and utilize the **causal mechanisms** that generate them. While adversarial models treat non-stationarity as arbitrary or even adversarial, our framework explains it as the result of structured causal processes unfolding over time. This distinction allows our method to reason about how and why rewards change and to plan interventions accordingly. We mentioned this in `Appendix F`.
>
> In particular, our framework enables **non-myopic planning** by identifying early interventions that have delayed effects via temporal causal paths. It also supports **mechanism-based explanations** of reward shifts (e.g., through transition edges in the causal graph), rather than relying solely on statistical drift. This structure is especially relevant in domains where intervention planning and interpretability are critical, such as healthcare, education, and policy-making—areas where blindly adapting to reward trends without causal understanding may be insufficient or even harmful.
>
> Although adversarial bandits offer sharper theoretical guarantees in worst-case settings, they typically do not incorporate causal knowledge and cannot inform **targeted interventions** grounded in the system’s structure. In contrast, our approach builds on the assumption that such structure (e.g., causal graphs) is either known or hypothesized, and leverages it to constrain the space of intervention sequences. Once this causal abstraction is in place—e.g., via the POMIS$^+$ sets—**any bandit algorithm**, including those developed for adversarial settings, can be used on top. In this sense, our contribution serves as a **causal decision-theoretic layer** that enhances interpretability and long-term planning, rather than replacing regret-minimizing solvers.
>
> We will revise the manuscript to better highlight this distinction and the complementary nature of our framework with respect to existing bandit approaches.

---

> > ### Comment · Reviewer_4V3R · 2025-08-03
> >
> > I thank the authors for their detailed and high-quality response.
> >
> > I am largely convinced by the additional clarifications they provided regarding the motivation and relevance of the proposed framework. In particular, I agree that applications involving sensitive policy-making require an understanding of complex causal structures, rather than simply adapting to observed payoff variations.
> >
> > While I still find the proposed framework somewhat overly complex, and the results not particularly sharp, I recognize the potential value in laying the foundations for a new line of theoretical research. I appreciate the authors' effort to position their work as a step toward broader developments within the community.
> >
> > In light of the authors’ commitment to better motivate their framework and improve the presentation—especially when introducing new concepts—I have decided to increase my score from 3 to 4.

---

### Official Review · Reviewer_KTWZ · 2025-07-01

**Clarity:** 3
**Significance:** 3
**Originality:** 3
**Rating:** 5
**Confidence:** 4

**Summary:**

This paper introduces a non-stationary causal bandit framework that models evolving causal mechanisms over time. The authors propose POMIS+, a method to identify interventions that optimize both immediate and long-term rewards by analyzing temporal propagation effects. . Experiments show improved performance over myopic baselines in dynamic environments.

**Questions:**

I have some questions for authors:

Assumption 3.1 implies that transition edges may exist only between consecutive time slices (i.e., first-order). I am not exactly sure why this assumption is needed in the first place—why POMIS+ can still be used when this assumption does not hold, just by increasing the length of the causal diagram window. If otherwise, can you explain what is the actual reason behind this assumption other than reducing computational issues?

Also, regarding Assumption 3.3, why must information propagation occur at every time step? What if this does not hold for consecutive time steps where information propagation does not occur? Will the proposed method fail to handle this case, or will it still work?

How scalable is the proposed algorithm? For bigger causal graphs with multiple sources of information propagation across time steps, will it be able to effectively identify the POMIS+?

**Ethical Concerns:**

["NO or VERY MINOR ethics concerns only"]

**Final Justification:**

I thank the reviewer for the detailed response to my questions. Including some discussion from the response will improve the manuscript. I will keep score for the paper.

**Limitations:**

yes

**Paper Formatting Concerns:**

N.A.

**Quality:**

3

**Strengths And Weaknesses:**

Summary of Contributions: The paper has three main contributions:
(1) a novel framework integrating structural causal models with non-stationary bandits for principled modeling of evolving causal mechanisms; (2) the POMIS+ method, which captures the existence of variables that contribute to maximizing both immediate and long-term rewards; and (3) comprehensive experimental validation demonstrating POMIS+'s superior performance in capturing long-term effects compared to myopic POMIS baselines across three distinct test settings.

Critique:
The main limitations of the proposed work are the assumptions. The identical time-slice structure assumption (Assumption 3.2) may restrict real-world applicability. Furthermore, Assumption 3.1's constraint that transition edges exist only between consecutive time slices could be overly restrictive in practice. The paper would benefit from a more thorough justification of these assumptions, particularly regarding why Assumption 3.1 is necessary.

Overall Assessment:
Despite these limitations, the paper is technically solid and makes several meaningful contributions. I therefore recommend acceptance.

---

> ### Author Rebuttal · Authors · 2025-07-31
>
> > Assumption 3.2 may restrict real-world applicability.
>
> Thank you very much for bringing this to our attention. We agree that Assumption 3.2 may indeed limit applicability in certain real-world scenarios. For example, in domains such as healthcare or industrial control systems, the underlying causal graph  $\mathcal{G}\bigl[\mathbf{V}_t\bigr]$ may evolve over time due to policy changes or system reconfigurations—such as changes in causal relationships between variables.
>
> In fact, our method remains valid even when Assumption 3.2 is relaxed, allowing for such structural changes. This is because identifying POMIS$^+$ at each time step $t$ only requires access to the ancestral graph of $Y_t$ — i.e., $\mathcal{G}\bigl[An(Y_t)\bigr]$. Therefore, our proposed Algorithm 1 ("Computing all intervention sequences") remains sound and complete even when this assumption is relaxed.
>
> The reason we initially adopted this assumption was that it aligns with a natural form of traditional non-stationary bandit in our setting—where causal dependencies may change over time *through the presence of transition edges*, which encode the propagation of information across time. This modeling choice aligns with prior work, such as Aglietti et al. [1], which makes a same Assumption 1.1 in the context of dynamic causal optimization in the time series manner.
>
> We discuss this assumption further in `Appendix L`. Due to space limitations, we were unable to include this discussion in the main body, but we will be sure to incorporate it in the camera-ready version to improve clarity.
>
> [1] Aglietti, V., Dhir, N., González, J., & Damoulas, T. (2021). Dynamic causal Bayesian optimization. *Advances in Neural Information Processing Systems*, *34*, 10549-10560.
>
> > Assumption 3.1's constraint that transition edges exist only between consecutive time slices could be overly restrictive in practice.
>
> > The paper would benefit from a more thorough justification of these assumptions, particularly regarding why Assumption 3.1 is necessary.
>
> Thank you for raising this important point. You are right that Assumption 3.1 “Time-slice Markov'”—that transition edges exist only between consecutive time slices—could be restrictive in real-world scenarios and our proposed method can still be used if the assumption does not hold.
>
> In traditional structural causal models (SCMs), temporal behavior is captured via topological ordering (i.e., causal order) over nodes in the graph, which we might interpret as a notion of time. This is a traditional Markovian property. However, in our setting—non-stationary structural causal bandits—the agent operates in a repeated decision-making scenario where each time step corresponds to a separate episode. Therefore, a purely topological interpretation of time is insufficient. We require a more explicit representation of temporal transitions to account for how causal influence propagates across real time steps.
>
> This is why we formally introduce the notion of “**time-slice Markov”** as Assumption 3.1—where transitions are constrained to flow between adjacent slices (i.e. first-order Markov). It serves as a practical way to simplify temporal modeling while preserving clarity of the time series. We also define its counterpart, “**semi time-slice Markov”**, in `Appendix G` ”Semi Time-Slice Markovian Non-Stationary SCM-MAB”, inspired by the classical notion of semi-Markovian graphs (i.e., DAGs with latent confounding represented by bidirected edges). Initially, we restricted “semi time-slice Markov” to cases involving latent confounding between two time steps—i.e., Dynamic Unobserved Confounders (DUC).
>
> However, your insightful question made us realize that **any violation of the first-order transition assumption—including observed, long-range dependencies across time—should also be captured under the broader semi-Markov category**. We appreciate this clarification and will revise the appendix to include these cases, while ensuring that the main text clearly states that our proposed methods remain valid under the relaxed Assumption 3.1 (Semi Time-slice Markov). This clarification helps reduce potential confusion about the role of Assumption 3.1 and makes the paper more precise and complete—thank you again for your valuable feedback.
>
> ***Questions:***
> > Assumption 3.1 implies that transition edges may exist only between consecutive time slices (i.e., first-order).  I am not exactly sure why this assumption is needed in the first place—why POMIS$^+$ can still be used when this assumption does not hold, just by increasing the length of the causal diagram window. If otherwise, can you explain what is the actual reason behind this assumption other than reducing computational issues?
>
> Thank you for your thoughtful follow-up question. As we discussed in our response to your earlier comment on Assumption 3.1, the constraint that transition edges exist only between adjacent time slices (i.e., first-order) is not required for the correctness of our method. POMIS$^+$ can still be applied when this assumption does not hold, including in cases involving longer-range or non-local temporal dependencies such as n-th order cases.
>
> The reason we introduce Assumption 3.1 is not to enforce a limitation, but to provide an interpretable *temporal* abstraction. In our non-stationary bandit setting, episodes unfold in real time, and thus we explicitly model temporal transitions across slices. By assuming a first-order structure, we simplify both exposition and implementation without losing generality. As clarified above, our method remains valid under a broader “semi time-slice Markov” condition, which we will expand on in `Appendix G` to include cases beyond the first-order. We appreciate your comment, as it helped us better articulate the generality of our assumptions.
>
> > Regarding Assumption 3.3, why must information propagation occur at every time step?What if this does not hold for consecutive time steps where information propagation does not occur? Will the proposed method fail to handle this case, or will it still work?
>
> Thank you very much for your accurate reading of our paper and for raising this insightful question.
>
> To directly address your concern: **our proposed method still works even if Assumption 3.3 does not hold**. However, when information propagation does not occur across certain time steps (i.e., the transition edges are missing between some time-slices $\mathbf{V}_{t}$ and $\mathbf{V}_s$ where $t \neq s$), the temporal structure becomes disconnected, and we consider the environment as the separated non-stationary SCM-MABs. If the information propagation does not occur across all time steps , we may have stationary graphs which illustrated in Lee and Barenboim [2]. In such cases, we apply our method **separately** to each connected segment of the causal graph.
>
> For instance, consider a scenario where $t = \lbrace 1,2,3,4 \rbrace$ and the structure is such that $\mathbf{V}_{1} \rightarrow \mathbf{V}_2$, there are **no** transition edges between $\mathbf{V}_2$ and $\mathbf{V}_3$, and then $\mathbf{V}_3 \rightarrow \mathbf{V}_4$. In this case, our algorithm can still be used by running it independently on each segment: $\mathcal{G} \bigl[ \mathbf{V}_1 \cup \mathbf{V}_2 \bigr]$ and $\mathcal{G}\bigl[ \mathbf{V}_3 \cup \mathbf{V}_4 \bigr]$. Conceptually, this corresponds to handling **two independent non-stationary decision-making problems**.
>
> While a user of the algorithm could reasonably manage such cases in practice, we introduced Assumption 3.3 to define a *normal-form* NS-SCM-MAB setting. We acknowledge that this may have caused confusion, and we appreciate your feedback—it will help us clarify the scope and flexibility of our method in the final version. We will make sure to incorporate this clarification into the main text to prevent the confusion occurred from Assumption 3.3.
>
> [2] Lee, S., & Bareinboim, E. (2018). Structural causal bandits: Where to intervene?. *Advances in neural information processing systems*, *31*.
>
> > How scalable is the proposed algorithm? For bigger causal graphs with multiple sources of information propagation across time steps, will it be able to effectively identify the POMIS$^+$?
>
> Thank you for the excellent question. We agree that the scalability of our algorithm is inherently tied to the complexity of the causal graph, particularly when there are multiple sources of information propagation across time steps. Since our method is based on **graphical criteria alone**, the size of the graph—and in particular, the number of nodes and transition edges—can impact the computational cost of identifying POMIS$^+$ sets.
>
> However, we emphasize that our algorithm is designed to operate as an **offline preprocessing step**, performed prior to any online decision-making or reward learning. This separation allows the intervention search to be done once, and reused thereafter.
>
> Moreover, in many real-world applications, the temporal causal graph exhibits **repeated or modular structure**—for instance, environments where the graph evolves periodically over a fixed time window $\tau$ (e.g., weekly behavior cycles or seasonal effects). In such cases, we only need to compute optimal intervention sets within this repeating window. Once identified, the same POMIS$^+$ sequence (or a variant) can be **reused across cycles**, substantially reducing the effective computation cost in practice. We will revise the paper to make this practical consideration clearer and to highlight how recurring structural patterns can mitigate computational challenges in real settings.

---

> > ### Comment · Reviewer_KTWZ · 2025-08-02
> > **Re.**
> >
> > I thank the reviewer for the detailed response to my questions. Including some discussion from the response will improve the manuscript. I will keep score for the paper.

---

> > > ### Author Response · Authors · 2025-08-06
> > >
> > > Dear Reviewer `KTWZ`,
> > >
> > > Thank you for your response to our rebuttal. We will make sure to incorporate the points discussed, especially those related to our assumptions, in the final version of the paper.
> > >
> > > In particular, we agree that your suggestion regarding Assumption 3.1—namely, broadening the definition of “semi time-slice Markov” to also include observed long-range temporal dependencies—is a valuable insight. We believe that explicitly incorporating this clarification into both the main text and `Appendix G` will greatly enhance the clarity and completeness of our theoretical framework.
> > >
> > > If you have any further questions or concerns about our work, please don’t hesitate to reach out—we would be happy to continue the discussion.
> > >
> > > Sincerely,
> > >
> > > Authors of submission 27658

---

### Official Review · Reviewer_Jv4n · 2025-07-02

**Clarity:** 3
**Significance:** 2
**Originality:** 2
**Rating:** 3
**Confidence:** 3

**Summary:**

The paper studies a setting of the multi-armed bandit (MAB) problem with causal structure, where the environment is non-stationary, unlike in previous work. The author formally introduces the non-stationary SCM-MAB framework using a graphical representation. The main contribution of the paper is to reduce the size of the action space by removing the actions that could never be optimal. To achieve this, the authors extend the classic result of Lee and Bareinboim (2018) to their setting.

**Questions:**

1- Could you elaborate on the points noted in the section above? Specifically, what are the new technical contributions and challenges that your work addresses?

2- The computational complexity of Algorithm 1 is exponential in the number of nodes. For large-scale graphs, this makes the algorithm impractical. How flexible are your results for extension to parametric SCMs (e.g., linear models)?

**Ethical Concerns:**

["NO or VERY MINOR ethics concerns only"]

**Final Justification:**

Overall, I agree that the problem and its motivation are important. The extension from non-stationary bandits to causal bandits is also novel. However, I'm still not convinced that the only practical use of the causal graph is to prune a subset of actions offline, after which a standard bandit algorithm is applied. That’s why I still lean toward rejection, although I have increased my score from 2 to 3.

**Limitations:**

Yes.

**Quality:**

3

**Strengths And Weaknesses:**

The paper addresses an interesting problem in the causal bandit literature and provides solid theoretical analysis. However, I am concerned about the novelty and contribution of the work. The main contribution appears to be identifying the best possible action by extending results from Lee and Bareinboim (2018). It does not seem that this extension is particularly challenging.
Furthermore, the paper provides neither a worst-case nor an instance-optimal regret guarantee with respect to the SCM. Of course, if we treat POMIS⁺ as independent arms, we could bound the regret using the classic bounds of KL-UCB or Thompson sampling, but there is no discussion of the optimality of this approach.

---

> ### Author Rebuttal · Authors · 2025-07-31
>
> > I am concerned about the novelty and contribution of the work. The main contribution appears to be identifying the best possible action by extending results from Lee and Bareinboim (2018). It does not seem that this extension is particularly challenging.
>
> > Could you elaborate on the points noted in the section above? Specifically, what are the new technical contributions and challenges that your work addresses?
>
> Thank you for the thoughtful comment. We would like to take this opportunity to clarify how our work differs from prior approaches. Although inspired by the structural causal bandit framework of [1], our approach constitutes distinct challenges going beyond a mere extension.
>
> - Original POMIS  [1]: Identifies **a possibly-optimal set of variables** to intervene on at a single time step, assuming a stationary causal model where the causal structure and reward distribution remain fixed over time.
> - Our NS-SCM-MAB framework: Identifies a possibly-optimal **sequence of intervention sets** across multiple time steps, under a non-stationary causal model where the causal structure and reward mechanisms evolve over time.
>
> To aid clarity, we begin by explicitly highlighting the limitations of the previous framework [1] before introducing our own. These limitations are clearly illustrated in the motivating example provided in `Section 3.1`. As shown in **Figure 4(a)**, the existing method **fails to select the essential variable** $X_1$, leading to a policy that **fails to converge**, even as the number of trials increases. This demonstrates that neglecting long-term causal dependencies can cause interventions to fail and result in **substantial performance degradation**.
>
> The core challenge in the NS-SCM-MAB setting is not simply increased difficulty—it requires more granular, graph-theoretically grounded control over how interventions impact future outcomes. To preserve optimality across time-indexed rewards $Y_t$, we introduce the concept of POMIS$^+$, which explicitly incorporates both immediate and temporally propagated effects. To support this, we develop **two new graphical tools**, **IB$^+$ and QIB**, and prove that their union is sufficient to satisfy the POMIS$^+$ condition (Proposition 5.1).
>
> A further challenge lies in the need to **enumerate all valid intervention sequences**, not just identify a POMIS$^+$ set for a single $Y_t$. This motivates our second major contribution: a recursive algorithm, grounded in Proposition 5.1, that is **provably sound and complete** in generating all admissible sequences composed of POMIS$^+$ sets. The level of structural and combinatorial rigor required here goes significantly beyond what prior approaches have attempted.
>
> To our knowledge, **no existing method**—including those developed for dynamic or non-stationary bandit problems—provides a principled framework for **characterizing or computing optimal intervention sequences** under evolving causal structures. Addressing this challenge necessitates a **fundamentally new approach**, encompassing novel theoretical results (e.g., Theorem 4.1), new graphical tools, and a recursive construction method (`Section 5`)—all of which represent substantial departures from previous work.
>
> While our approach is inspired by prior research on causal bandits, we now recognize the importance of **more clearly articulating** how our formulation advances those ideas—particularly with regard to modeling assumptions, temporal graph structure, and the scope of intervention planning. We appreciate your suggestion, and we will ensure that these distinctions and contributions are more prominently highlighted in the revised manuscript.
>
> [1] Lee, S., & Bareinboim, E. (2018). Structural causal bandits: Where to intervene?. Advances in neural information processing systems, 31.
>
> > Furthermore, the paper provides neither a worst-case nor an instance-optimal regret guarantee with respect to the SCM.
>
> > Of course, if we treat POMIS$^+$ as independent arms, we could bound the regret using the classic bounds of KL-UCB or Thompson sampling, but there is no discussion of the optimality of this approach.
>
> If one aims to minimize regret while accounting for dependencies among arms, it would require modeling the underlying SCM in its canonical form—i.e., representing the exogenous variables and their structural mappings [2]. However, this is not our primarily concern.
> Our goal is not to exploit inter-arm statistical dependencies for regret minimization, but rather to leverage structural properties of the causal diagram to reduce the intervention space. By identifying POMIS$^+$ sets via IB$^+$ and QIB from the observed causal graph $\mathcal{G}$, we propose a principled, SCM and causal graph based method for pruning the action sequence space prior to any bandit algorithm.
>
> The regret behavior of downstream algorithms can then be analyzed using existing regret bounds, applied to the reduced action space defined by POMIS$^+$. This **modular design** is a key advantage of our framework: rather than modifying the exploration procedure itself, it offers a **causally grounded narrowing of the intervention space** that can be used in conjunction with any standard bandit algorithm. While this does not alter the asymptotic regret class, the practical regret may be significantly lower—since regret bounds often scale with the size of the action set, and POMIS$^+$ helps restrict it meaningfully.
> We will revise the manuscript to clarify this distinction and avoid confusion about the intended scope of our contribution.
>
> [2] Zhang, J., Tian, J., & Bareinboim, E. (2022). Partial counterfactual identification from observational and experimental data. In *International Conference on Machine Learning* (pp. 26548-26558). PMLR.
>
> > The computational complexity of Algorithm 1 is exponential in the number of nodes. For large-scale graphs, this makes the algorithm impractical. How flexible are your results for extension to parametric SCMs (e.g., linear models)?
>
> We appreciate the concern regarding scalability, and we would like to take this opportunity to clarify the scope and practicality of our approach.
>
> Our method is designed as an **offline phase** for structural causal reasoning, performed prior to any bandit algorithm or online reward learning. The primary goal of Algorithm 1 is to identify possibly optimal intervention sequences using **purely graphical criteria**, without requiring any knowledge of the underlying functional mechanisms or noise distributions of the SCM.
>
> This is made possible because our algorithm builds on graphical tools such as **do-calculus**, IB$^+$, and QIB, which operate entirely at the level of the causal graph $\mathcal{G}$—a structural abstraction of the SCM. In other words, intervention sets are selected based on **causal dependencies encoded in** $\mathcal{G}$, not by computing interventional distributions such as $\mathbb{E}\bigl[Y \mid \operatorname{do}(\mathbf{x})\bigr]$.
>
> Therefore, even when the SCM is assumed to be linear, **the underlying causal graph typically remains unchanged**. As a result, the complexity of enumerating intervention sequences remains exponential, regardless of whether the SCM is linear or nonlinear. That is, the computational challenge arises from the size of the search space induced by the causal structure—not from the complexity of evaluating parametric models.
>
> Importantly, the graph-based reasoning we adopt is highly **practical and general**—especially in domains where full parameterization of the SCM is unavailable or hard to estimate. For example, while it may be difficult to precisely learn structural equations, the causal graph is often more readily obtainable (e.g., via data-driven causal discovery) or can be hypothesized based on domain knowledge. In such settings, our method offers a robust and assumption-light solution that avoids reliance on strong parametric forms.
>
> In practice, there are also opportunities to reduce the computational burden by leveraging **repeated graph structures**. For example, in many real-world environments, the underlying causal diagram exhibits **periodically repeating non-stationary patterns** over a fixed time window $\tau$. This reflects settings where structural shifts occur cyclically—such as seasonal changes in recommendation systems or periodic maintenance cycles in industrial processes.
>
> In such cases, once an optimal intervention sequence is computed for the first $\tau$ time steps, the same sequence (or its variants) can be reused in subsequent cycles. This makes the **effective time complexity depend only on** $\tau$, rather than the total episode length $T$. Although the complexity remains exponential in $\tau$, the reduction in effective horizon can significantly improve scalability in practice.
>
> We will incorporate this clarification into the revision to reflect how structure-level regularities—common in real-world applications—can help mitigate worst-case complexity while retaining theoretical guarantees.

---

> > ### Author Response · Authors · 2025-08-06
> >
> > Dear Reviewer `Jv4n`,
> >
> > Thank you very much for taking the time to review our paper. We sincerely appreciate your thoughtful feedback and the important points you raised.
> >
> > The main concerns highlighted in your review pertain to:
> >
> > **(1)** the novelty and technical contribution of our work beyond Lee & Bareinboim (2018),
> >
> > **(2)** the lack of explicit regret analysis and how our approach relates to classical regret minimization, and
> >
> > **(3)** the computational complexity of our algorithm and its scalability to larger graphs or parametric SCMs (i.e., linear SCM).
> >
> > We have thoroughly addressed each of these concerns in our rebuttal. Specifically,
> >
> > - We clarified how our framework moves beyond single-step causal bandits by introducing a temporally structured, non-stationary setting that requires identifying *sequences* of interventions.
> > - To do so, we developed new graphical tools (IB$^+$ and QIB), established new theoretical results (e.g., Proposition 5.1 and Theorem 4.1), and designed a recursive algorithm that is both sound and complete under the NS-SCM-MAB setting.
> > - We also explained how our approach is designed as a structural, offline intervention-planning method—complementary to existing bandit algorithms—whose practical regret can be improved via action-space pruning, even if it does not directly alter regret class.
> > - Finally, we discussed how structural regularities (e.g., $\tau$ periodic causal graphs) in real-world applications can mitigate the algorithm’s worst-case complexity.
> >
> > If there are any points that remain unresolved, we’d be glad to address them **in more technical detail**. We are genuinely looking forward to the possibility of a deeper conversation, as we believe your feedback could meaningfully improve our work.
> >
> > Sincerely,
> >
> > Authors of Submission 27658

---

> > ### Comment · Reviewer_Jv4n · 2025-08-06
> >
> > Thanks for the response. Overall, I agree that the problem and its motivation are important. The extension from non-stationary bandits to causal bandits is also novel. However, I'm still not convinced that the only practical use of the causal graph is to prune a subset of actions offline, after which a standard bandit algorithm is applied. That’s why I still lean toward rejection, although I have increased my score from 2 to 3.

---

### Official Review · Reviewer_k47E · 2025-07-03

**Clarity:** 2
**Significance:** 3
**Originality:** 2
**Rating:** 4
**Confidence:** 3

**Summary:**

The paper introduces NS-SCM-MAB, a non-stationary structural causal bandit framework that models evolving causal mechanisms and captures how interventions affect both immediate and future rewards. It extends prior work by incorporating temporal dynamics into causal graphs and proposes a method for identifying the set of interventions that includes the best possible sequence of actions. The authors develop an algorithm to find such interventions and show, both theoretically and empirically, that it outperforms myopic strategies in dynamic environments.

**Questions:**

- Definition 3.1 may require clarification: shouldn't it depend on whether $Y_t = Y_{t'}$ (i.e., whether the reward corresponds to the same node across time)? Even in stationary environments, if $Y_t \neq Y_{t'}$, the stated equality may not hold.

- Can authors explain the necessity of Assumption 3.3?

**Ethical Concerns:**

["NO or VERY MINOR ethics concerns only"]

**Final Justification:**

Thank you to the authors for their detailed responses, which have addressed part of my concerns.

I am now convinced of the novelty of the paper. However, I remain unconvinced regarding the second issue. Similar to the work of Lee & Bareinboim (2018), this paper focuses on identifying possibly optimal intervention sets, that is, determining sets of nodes that can be removed from the action space a priori, without any sampling or interaction with the environment. In my view, this approach is somewhat incomplete, as it does not address whether such pre-processing is the only viable use of the causal structure. An important unaddressed question is whether incorporating causal information and extra observations during the learning process, over these possibly optimal sets, could lead to improved regret bounds.

That said, given that this is the first work to study non-stationary networks in causal bandits, I see this as a promising direction for future work. I will therefore increase my score to 4.

**Limitations:**

Refer to strengths and weaknesses.

**Paper Formatting Concerns:**

No formatting concerns.

**Quality:**

3

**Strengths And Weaknesses:**

**Strengths:**

- The paper addresses an important and realistic setting, as causal mechanisms in the real world are often non-stationary. This aspect has not been explored in prior work.
- The theoretical foundation is solid; the authors show that their algorithm is both sound and complete.

**Weaknesses:**

- The novelty of the contribution is unclear. The proposed approach appears to be a natural extension of the framework by Lee and Bareinboim [2018] to non-stationary environments.
- There is no regret lower bound or discussion of optimal regret after identifying the set of POMIS$^+$. It is unclear whether a more efficient strategy than standard UCB could be applied over the POMIS$^+$ sets.
- Figure 6(b), which is referenced in lines 128 and 188, is missing from the paper.

---

> ### Author Rebuttal · Authors · 2025-07-31
>
> > The novelty of the contribution is unclear. The proposed approach appears to be a natural extension of the framework by Lee and Bareinboim [2018] to non-stationary environments.
>
> Thank you for the thoughtful comment. We would like to take this opportunity to clarify how our work differs from prior approaches. While our work is indeed inspired by the structural causal bandit framework of [1], our contribution departs significantly from theirs, both conceptually and technically.
>
> The key distinction lies in our treatment of **non-myopic intervention sequences** under temporal causal dynamics. The original POMIS framework identifies a set of possibly-optimal interventions under a stationary causal model. In contrast, our framework—NS-SCM-MAB—explicitly models evolving causal mechanisms over time and aims to identify a set of all possibly-optimal ***sequences* of interventions** spanning multiple time steps.
>
> The limitation of existing method is clearly illustrated in our motivating example (see `Section 3.1`). As shown in **Fig. 4(a)**, **the existing method fails to select the crucial variable** $X_1$, resulting in a policy that does **not converge at all**, even with increased trials. This example highlights that **ignoring long-term causal dependencies can lead to intervention failures with severe performance consequences**.
>
> The underlying challenge in the NS-SCM-MAB setting is not merely harder—it demands finer-grained and graph-theoretically precise control over how interventions influence future outcomes. To conserve optimality for each reward $Y_t$ , we introduce the notion of POMIS$^+$, which accounts for both immediate and temporally propagating effects. We further develop **two novel graphical tools**—**IB$^+$ and QIB**—and prove that their union satisfies the POMIS$^+$ condition (Proposition 5.1).
>
> An additional difficulty lies in the need to **identify all such valid intervention sequences**, not just identifying POMIS$^+$ for a single $Y_t$. This motivates our second contribution: a recursive algorithm that, grounded in Proposition 5.1, is theoretically **sound and complete** in enumerating every valid sequence composed of POMIS$^+$ sets. This level of combinatorial and structural rigor goes well beyond what existing methods achieve.
>
> To the best of our knowledge, **no existing method**—including those developed for dynamic or non-stationary bandits—offers a principled way to **characterize or compute structurally optimal intervention sequences** in the presence of evolving causal dependencies. Addressing this challenge requires a **fundamentally new framework**, including new theoretical results (e.g., Theorem 4.1), new graphical tools, and a new recursive algorithm (see `Section 5`)—all of which go significantly beyond the scope of prior work.
>
> Therefore, while our work builds on earlier causal bandit literature, we now recognize the importance of **more explicitly articulating** how our formulation generalizes prior frameworks—particularly in terms of modeling assumptions, graphical structure, and the nature of intervention sequences. We thank you for prompting this clarification, and we will ensure that these distinctions and contributions are more clearly emphasized in the revised version of the paper.
>
> [1] Lee, S., & Bareinboim, E. (2018). Structural causal bandits: Where to intervene?. *Advances in neural information processing systems*, *31*.
>
> > There is no regret lower bound or discussion of optimal regret after identifying the set of POMIS$^+$. It is unclear whether a more efficient strategy than standard UCB could be applied over the POMIS$^+$ sets.
>
> Thank you for raising this point. We respectfully clarify that the focus of our work lies not in designing a new bandit algorithm, but in integrating causal structure into non-stationary bandit problems to reduce the set of intervention candidates through structural causal reasoning. Specifically, our framework leverages SCM and the causal diagram to identify POMIS$^+$ sequences, which represent minimal sets of possibly-optimal interventions over time. By doing so, we aim to **narrow the action sequence space** in a principled way based on causal structure (i.e., off-line pre-processing), rather than improving regret through algorithmic innovation.
>
> Once the POMIS$^+$ sets are identified, any standard bandit algorithm—such as KL-UCB or Thompson Sampling—can be applied to explore over these reduced sets. **The regret analysis of such algorithms would then follow from existing regret bounds**, conditioned on the restricted arm space defined by POMIS$^+$. We believe this modularity is a strength of our framework: it complements existing bandit strategies by providing a **causally informed reduction of the decision space**, rather than replacing the exploration mechanism itself. Although the asymptotic regret class remains unchanged, reducing the number of arms through POMIS$^+$ can lead to smaller regret in practice, as regret bounds typically scale with the number of arms.
>
> We appreciate the opportunity to clarify this distinction and will revise the manuscript to make this point more explicit.
>
> > Figure 6(b), which is referenced in lines 128 and 188, is missing from the paper.
>
> Thank you for pointing this out. The figure reference should have been to Fig. 7(b), which depicts a scenario in which the causal graph over time remains identical across time steps, without any transition edges—i.e., the stationary SCM-MAB case. This serves as the foundation upon which we define our non-stationary setting. We appreciate your attention to detail and will ensure that all such referencing issues are corrected in the final version.
>
> ***Questions:***
>
> > Definition 3.1 (Non-stationary SCM-MAB) may require clarification: shouldn't it depend on whether $Y_{t}=Y_{t′}$ (i.e., whether the reward corresponds to the same node across time)? Even in stationary environments, if $Y_{t}\neq Y_{t′}$, the stated equality may not hold.
>
> We thank the reviewer for the comment. We would like to clarify that Definition 3.1 does not rely on whether $Y_t = Y_{t'}$ in a literal variable sense. In our framework, $Y_t$ and $Y_{t'}$ are **time-indexed reward variables** that represent the same underlying outcome measured at different time steps (e.g., blood sugar level at $t=1$ vs. $t=5$). This is conceptually distinct from comparing entirely different variables (e.g., blood sugar vs. LDL cholesterol), which would require a different notation or modeling framework altogether—such as defining a multi-dimensional outcome vector or using separate variables like $Y_t$ and $W_{t'}$.
>
> Our setting assumes a consistent reward semantics across time, and non-stationarity is defined in terms of shifts in the **distribution** of this time-indexed reward under identical interventions. Thus, Definition 3.1 remains valid even when $Y_t \neq Y_{t'}$ as variable names—because what matters is whether their conditional distributions under the same intervention remain invariant (e.g., $P(LDL_1 \mid \operatorname{do(statin_1)}) = P(LDL_5 \mid \operatorname{do(statin_5)})$.
>
> Scenarios involving fundamentally different outcomes across time are outside the scope of our current formulation.
>
> > Can authors explain the necessity of Assumption 3.3?
>
> Thank you for this thoughtful question. We introduced Assumption 3.3 to define **a normal-form NS-SCM-MAB setting**, where non-stationarity arises due to continuous information propagation across time. This assumption ensures that the temporal causal model is connected from $t = 1$ to $t = T$, enabling a clean theoretical characterization of how information and intervention effects evolve over time.
>
> This structural connectivity serves as a necessary formal prerequisite for the theoretical and conceptual formulation of temporal POMIS$^+$ sets, which are designed to capture how early interventions influence future rewards across time. Without this assumption, the temporal graph may decompose into disconnected components, making it unclear whether a coherent intervention sequence can or should be optimized over the entire horizon.
>
> However, we emphasize that our method remains valid even when Assumption 3.3 does not hold. In such cases, the environment effectively reduces to a set of **disconnected subgraphs**, each of which can be handled independently using our proposed approach. This corresponds to **solving separate causal bandit problems** **over each segment of the time horizon** where information is locally propagated.
>
> Therefore, Assumption 3.3 rather serves to define the expressive setting in which our theory and algorithm can be jointly applied over the entire time horizon. We will revise the text to make this intention clearer and to highlight the flexibility of our method beyond this assumption. We hope this response addresses your concern. Should you have any further questions or require additional clarification, we would be glad to elaborate.

---

> > ### Author Response · Authors · 2025-08-06
> >
> > Dear Reviewer `k47E`,
> >
> > Thank you again for your thoughtful and constructive review. We truly value your careful review and the key issues you highlighted. Your main concerns focused on
> >
> > **(1)** whether the contribution goes beyond the structural causal bandit framework of Lee & Bareinboim (2018),
> >
> > **(2)** the lack of regret bounds or discussion of optimal regret strategies,
> >
> > **(3)** and clarification regarding core definitions such as Definition 3.1 and Assumption 3.3.
> >
> > In our rebuttal, we have carefully addressed each of these points in detail. Specifically, we clarified:
> >
> > - How our work departs from prior frameworks by introducing a new graphical formulation that captures evolving causal dependencies over time, an intervention concept (POMIS$^+$), which accounts for both immediate and temporally propagating effects.
> > - How we propose a recursive algorithm that is both sound and complete in enumerating valid **intervention** **sequences**. Our notions of POMIS$^+$, IB$^+$, and QIB allow for the enumeration of all valid intervention sequences, backed by theoretical guarantees of completeness and soundness.
> > - Regarding regret, we emphasized that our method reduces the candidate intervention space using causal structure, and that any standard bandit algorithm can then be applied over this reduced space.
> > - Lastly, we elaborated on the semantics of time-indexed variables and the structural role of Assumption 3.3, and will revise the final version accordingly.
> >
> > We truly appreciate your engagement and thoughtful feedback. We sincerely believe that your comments will guide meaningful improvements in the final version.
> >
> > If you have any further questions, we would be glad to address them.
> >
> > Sincerely,
> >
> > Authors of Submission 27658

---

> > > ### Comment · Reviewer_k47E · 2025-08-07
> > >
> > > Thank you to the authors for their detailed responses, which have addressed part of my concerns.
> > >
> > > I am now convinced of the novelty of the paper. However, I remain unconvinced regarding the second issue. Similar to the work of Lee & Bareinboim (2018), this paper focuses on identifying possibly optimal intervention sets, that is, determining sets of nodes that can be removed from the action space a priori, without any sampling or interaction with the environment. In my view, this approach is somewhat incomplete, as it does not address whether such pre-processing is the only viable use of the causal structure. An important unaddressed question is whether incorporating causal information and extra observations during the learning process, over these possibly optimal sets, could lead to improved regret bounds.
> > >
> > > That said, given that this is the first work to study non-stationary networks in causal bandits, I see this as a promising direction for future work. I will therefore increase my score to 4.

---

### Official Review · Reviewer_EXe2 · 2025-07-03

**Clarity:** 2
**Significance:** 4
**Originality:** 3
**Rating:** 4
**Confidence:** 3

**Summary:**

This paper introduces a non-stationary structural causal bandit framework to address sequential decision-making in dynamic environments with evolving causal mechanisms. Unlike traditional static approaches, the proposed method uses temporal structural causal models to capture long-term effects of interventions. The authors develop graphical tools and assumptions to analyze intervention propagation over time and introduce POMIS+, an extension of minimal intervention sets that optimizes both immediate and long-term rewards. Empirical results show the framework outperforms myopic baselines.

**Questions:**

- If Assumption 3.1 is not satisfied, what would happen to the theory of optimal intervention set?

- What does the non-stationarity originate from? It seems that the non-stationarity originates from the reward distribution shift, and this shift may come from the shifting causal structure.


- It is not clear whether the proposed method allows the shifting causal structure in the model.
This paper makes Assumption 3.2, which constrains the causal structures of each time slice to be identical.
However, in lines 62 and 73, it seems that POMIS+ allows the underlying causal structure to change.
Then, which variables are allowed to induce the shifting causal structure?


- Since $X$ is the arm of the bandit, why intervening $Z$ in Equation 1? It is not clear what $Z$ is.

- In Section 3.1, it might be better to briefly introduce MUCT and IB before giving the motivating example.

**Ethical Concerns:**

["NO or VERY MINOR ethics concerns only"]

**Limitations:**

This method makes the assumptions of the first-order Markovian structure, an identical time slice, and the existence of a transition edge.

**Paper Formatting Concerns:**

NA.

**Quality:**

4

**Strengths And Weaknesses:**

# Strengths

- This paper studies an important problem, which extends existing stationary MAB to the non-stationary one. It uses the lens of SCM to characterize the temporal causes of reward changes.

- The authors develop the POMIS+ algorithm that considers the long-term outcome effect of the current intervention.

- The authors theoretically analyze the optimal intervention sets.

# Weaknesses

- It is not quite clear how the authors tackle the non-stationarity problem intuitively.

- In Figure 1, the definitions of $Z$ are lacking.

- Some minor typos. For example, in lines 128 and 188, there is no Fig. 6(b) in the paper or the Appendix.

---

> ### Author Rebuttal · Authors · 2025-07-31
>
> > It is not quite clear how the authors tackle the non-stationarity problem intuitively.
>
> Thank you for raising this important point. We understand that the intuitive motivation for how our method tackles non-stationarity may not have been sufficiently clear from the current presentation.
>
> Our work addresses non-stationarity that arises from evolving causal mechanisms in terms of **structural changes** (i.e., causal dependencies that vary across time) and thus distributional changes (i.e., shifts in the reward distribution $P(Y_t \mid \operatorname{do}(\mathbf{X}_t))$). Unlike standard SCM-MAB settings that assume stationarity and focus on short-term effects, our method is designed to capture **temporal dependencies** and **long-term consequences of interventions**.
>
> To make this explicit, our contributions can be summarized as follows:
>
> - **Modeling**: We formulate a non-stationary structural causal model (NS-SCM) where each time variables $\mathbf{V}_t$ has its own temporal model $\mathcal{M}_t$ and causal graph $\mathcal{G}\bigl[\mathbf{V}_t\bigr]$, and causal influence propagates via defined **transition edges** between two time steps $t, t’ \in \bigl[T\bigr]$.
> - **Methodology**: We introduce a strategy that identifies **non-myopic interventions** by tracking how causal influence flows across time using graphical reasoning.
>     - **POMIS**$^+$: We develop the concept of **POMIS**$^+$  to address a key limitation of the classical POMIS approach, which only identifies variables that are optimal for the *current* time step. In contrast, **POMIS**$^+$  **captures variables that are crucial for maximizing both immediate and future rewards** by explicitly modeling how interventions propagate over time through the causal graph. To this end, we define two components—**IB$^+$**, which tracks intervention-relevant variables across time, and **QIB**, which characterizes the time-local minimal sets conditioned on IB$^+$—and take the union of them to construct valid POMIS$^+$ sets. Also, we theorically proved $IB^+ \cup QIB$ is a POMIS$^+$ (Proposition 5.1). This enables our method to identify long-term intervention strategies that classical POMIS cannot capture.
>     - **Algorithmic Contribution**: We design a recursive algorithm that enumerates **all valid intervention sequences** composed solely of POMIS$^+$  sets, guaranteeing **soundness and completeness** under a dynamic causal setting. The algorithm reduces computational cost by pruning unnecessary intervention sequences, thereby avoiding redundant searches and enabling tractable computation even in long-horizon decision settings.
>
> This design enables our method to move beyond myopic, step-wise decision-making and to account for **how present actions affect future rewards** through the temporal causal structure.
>
> We will revise the introduction and Section 3 to make these intuitions clearer, and add a motivating example that illustrates how non-stationarity impacts the intervention choice. Thank you again for your helpful feedback.
>
> > In Figure 1, the definitions of $Z$ are lacking.
>
>  In lines 119–121, we define $\mathbf{X} \subseteq \mathbf{V} \setminus \lbrace Y\rbrace$ to be a **set of manipulative variables**, not a single variable. As clarified in `Section 2`, we use capital letters for individual variables and bold capital letters for sets of variables. For example, if the causal diagram includes variables $X$, $Y$, and $Z$, then possible choices for $\mathbf{X}$ (i.e., subsets of $\mathbf{V} \setminus \lbrace Y \rbrace$) include $\emptyset$, $\lbrace X \rbrace$, $\lbrace Z \rbrace$, and $\lbrace X, Z \rbrace$. Thus, $Z$ is one of the variables that can be included in the set $\mathbf{X}$. To avoid confusion, we will revise the notation and denote the set as $\mathbf{A}$ instead of $\mathbf{X}$.
>
> > Some minor typos. For example, in lines 128 and 188, there is no Fig. 6(b) in the paper or the Appendix.
>
> We apologize for the typo that may have caused confusion. The correct reference is to Fig. 7(b), which illustrates the case where the causal structure among time-indexed variables is identical across all time steps and contains no transition edges—i.e., the stationary SCM-MAB setting. We construct our non-stationary setting by building on this stationary foundation. We will correct this typo in the camera-ready version. Thank you for pointing out this typo.
>
> ***Questions:***
> > If Assumption 3.1 is not satisfied, what would happen to the theory of optimal intervention set?
>
> Thank you for this insightful question. As described in `Appendix G`, we generalize Assumption 3.1 by introducing a broader structural condition based on **semi time-slice Markov graphs**—which allow for temporal dependencies that are not strictly captured by direct transition edges between consecutive time slices. Specifically, we consider cases where variables across time are connected through **latent confounding**, such as Dynamic Unobserved Confounders (DUCs). This formulation allows for the possibility that variables from non-adjacent time steps may exhibit confounding effects, thereby relaxing the strict temporal adjacency required in Assumption 3.1.
>
> Importantly, our proposed framework and algorithm remain applicable under this generalization. The identification of optimal intervention sets, including POMIS$^+$, still relies on causal information encoded in the temporal graph—regardless of whether Assumption 3.1 is strictly satisfied. The key requirement is that the causal structure (including latent links) correctly reflects how information flows over time.
>
> We introduced Assumption 3.1 primarily to present a **clean and interpretable normal-form model** for the main exposition. We appreciate your question, and in the revision we will clarify in the main text that our theoretical results extend beyond this assumption via the structural conditions formalized in `Appendix G`.
>
> > What does the non-stationarity originate from? ... this shift may come from the shifting causal structure.
>
> We clarify that in our framework, **non-stationarity originates not merely from reward distribution shifts, but from structural changes in the underlying causal mechanisms over time**.
>
> Specifically, as shown in **Figure 1(a)**, the temporal causal graph includes **transition edges**—depicted in red—that connect variables from one time step (e.g., $X_t$, $Z_t$) to variables in the subsequent step (e.g., $X_{t'}$, $Z_{t'}$). These edges indicate that the value of future variables is not only determined by contemporaneous parents (within time $t'$), but also by prior variables from time $t$.
>
> In the corresponding SCM (Figure 1(b)), this is formalized by structural functions such as: $f_{z_{t'}} = U_{z_{t'}} \oplus x_{t'} \oplus x_t$
>  which show that $Z_{t'}$ depends explicitly on past values $x_t$ and $x_{t'}$. These variables then influence rewards like $Y_{t'}$.
>
> Thus, while the reward distribution may appear to shift over time (i.e., $P(Y_t \mid \operatorname{do}(X_t)) \neq P(Y_{t'} \mid \operatorname{do}(X_{t'}))$), this shift is **not arbitrary**. It is explained by **causal information propagating across time**, encoded by transition edges in the graph and captured through structural equations. In this sense, our framework does not treat non-stationarity as a black-box drift, but rather provides a **mechanism-based explanation** grounded in the evolving structure of the causal model.
>
> This distinction is crucial, as it enables principled reasoning about why rewards change—allowing for informed, temporally-aware interventions.
>
> > It is not clear whether the proposed method allows the shifting causal structure in the model. ... Then, which variables are allowed to induce the shifting causal structure?
>
> As we noted previously, our framework models non-stationarity not through changes in the causal structure within each time slice, but through causal information propagation across time steps via transition edges.
>
> To clarify your current question: the variables that induce shifting causal influence are those that serve as the **source nodes of transition edges**—namely, variables such as $X_t$ and $Z_t$ at time $t$, which have **outgoing edges pointing to variables at time** $t’$.
>
> As illustrated in **Figure 1(a)**, these variables have outgoing edges to future variables like $X_{t’}$ or $Z_{t’}$, introducing **time-varying dependencies**. For instance, $Z_{t’}$ is determined by a realized value of $X_t$ and $X_{t’}$ is set by a value of  $Z_t$ as shown in the SCM (Figure 1(b)).
>
> This means that although **each time slice shares the same internal structure** (as per Assumption 3.2), the **full temporal model exhibits structural evolution**, because the set of active parents for each variable grows across time via these inter-slice dependencies.
>
> So in short: the structure of the entire system shifts over time, and the **variables that induce this shift are precisely those that participate in transition edges—like** $X_t$ **and** $Z_t$**—whose influence spans across time steps**. This mechanism is what enables our framework to capture non-stationarity in a structured and interpretable manner.
>
> > Since X is the arm of the bandit, why intervening Z in Equation 1? It is not clear what Z is.
>
> We intended the set of possible arms of the bandit to be $\mathbf{V} \setminus \lbrace Y \rbrace$, which we denote as $\mathbf{Z}$ in line 197 when formulating Equation (1). As stated above, we will revise the notation and consistently refer to the set of possible arms as $\mathbf{A}$, instead of using any other symbol.
>
> > In Section 3.1, it might be better to briefly introduce MUCT and IB before giving the motivating example.
>
> Without familiarity with MUCT and IB, our work may be difficult to fully assess, even though we introduce these definitions in `Appendix B`. We will include them in the camera-ready version. Thank you for this suggestion.

---

> > ### Author Response · Authors · 2025-08-06
> >
> > Dear Reviewer `EXe2`,
> >
> > Thank you so much for your great efforts to review our work. We greatly appreciate your detailed feedback and the critical points you brought to our attention. Your major concerns are:
> >
> > **(1)** what our framework—non-stationarity in a graphical causal lens—originates from,
> >
> > **(2)** how we tackle this non-stationarity problem,
> >
> > **(3)** and what would happen to the theory of POMIS$^+$ if Assumption 3.1 is not satisfied.
> >
> > We have diligently addressed each of these concerns in detail throughout the rebuttal:
> >
> > - We clarified that non-stationarity arises from structural evolution over time—specifically, via transition edges that propagate causal influence across time steps, as shown in Figure 1. This structural view contrasts with black-box distributional drift and allows for principled, temporally-aware intervention planning.
> > - To tackle this, we introduced an algorithmic graphical approach that systematically tracks these inter-temporal dependencies. Our concepts of POMIS$^+$, IB$^+$, and QIB are all grounded in this mechanism, enabling us to **systematically enumerate all valid intervention sequences** that are relevant across time, with theoretical guarantees of soundness and completeness.
> > - Regarding Assumption 3.1, we highlighted in `Appendix G` that our theory extends to more general settings via a **semi time-slice Markov** formulation, which allows for non-local dependencies, including latent confounding across time steps. We clarified that the correctness of our framework—including the computation of POMIS$^+$ sequences—does not hinge on this assumption.
> > - Additionally, we responded to your other comments regarding unclear notation, definitions, and typo. These will all be reflected in the camera-ready version with careful revisions.
> >
> > If you have any remaining questions or further concerns, we would be more than happy to discuss them. We truly appreciate your thoughtful review and your engagement with our work.
> >
> > Sincerely,
> >
> > Authors of Submission 27658

---

> > > ### Comment · Reviewer_EXe2 · 2025-08-06
> > > **Official Comment by Reviewer EXe2**
> > >
> > > Thanks for the detailed response.
> > >
> > > I am still confused about the origin of the non-stationarity. From your explanations, I still think the proposed model is stationary, due to Assumption 3.2. Specifically,
> > >
> > > - i) In Figure 1 (a), previous variables (for example, X_{t-1}, Z_{t-1}, and Y_{t-1}) could also affect those at the current time step (i.e., X_{t}, Z_{t}, and Y_{t}), SO the SCM in Figure 1(b) seems not correct.
> > >
> > > - ii) With Assumption 3.2 and the corrected SCM, we find that both causal structure and causal effects remain identical, and thus this model is stationary.
> > >
> > > - iii) The sentence that you described, "because the set of active parents for each variable grows across time via these inter-slice dependencies.", is not correct, because in each time slice, the parents of each variable remain the same, and will not grow.
> > > BTW, what does "active" mean here?

---

> > > > ### Author Response · Authors · 2025-08-06
> > > >
> > > > Thank you for the detailed comments. We believe the confusion arises from the temporal models (as defined in Def 3.2) and the role of Assumption 3.2 in our framework. We would like to clarify the origin of non-stationarity and explain how our model departs from stationarity, even under Assumption 3.2.
> > > >
> > > > First, **Assumption 3.2** (Identical Time Slice) ensures that the *intra-slice* causal structure—i.e., the graph among variables $\mathbf{V}_t$ within a single time step—is the same across time. However, this does **not** imply that each SCM (i.e., each temporal model) is stationary. What introduces non-stationarity in our framework is the presence of **transition edges**—edges from variables at time $t$ to variables at time $t' > t$ (as shown in **Figure 1(a)**). These inter-slice edges connect identical time slices into a **temporally unrolled causal graph**, introducing time-dependent causal influences.
> > > >
> > > > More specifically:
> > > >
> > > > - **(i)** Regarding your comment about Figure 1(a) and (b): we respectfully clarify that Figure 1(b) depicts structural assignments consistent with Figure 1(a). For example, $f_{Z_{t'}} = U_{Z_{t'}} \oplus x_t \oplus x_{t'}$ directly reflects that $Z_{t'}$ receives causal inputs from both $X_{t'}$ (within-slice) and $X_t$ (cross-slice). There is no contradiction between the graph and the functions; rather, they together demonstrate how transition edges introduce new parents over time, which is the source of non-stationarity.
> > > > - **(ii)** Assumption 3.2 only constrains the **within-slice structure**, not **cross-slice dependencies**.
> > > > Perhaps, your main concern arises from the following: although the *local topology* of each time slice remains fixed (i.e., same two causal graphs $\mathcal{G}\bigl[\mathbf{V}_t\bigr] = \mathcal{G}\bigl[\mathbf{V}\_{t'}\bigr]$, where time $t < t'$), there exists an important underlying difference between two time slices. Specifically, the two temporal models $\mathcal{M}_t$ and $\mathcal{M}\_{t'}$ that generate these two causal graphs $\mathcal{G}\bigl[\mathbf{V}_t \bigr] \sim \mathcal{M}_t$ and $\mathcal{G}\bigl[\mathbf{V}\_{t'}\bigr] \sim \mathcal{M}\_{t'}$ may differ (i.e., $\mathcal{M}_t \neq \mathcal{M}\_{t'}$).
> > > > The disparity between $\mathcal{M}_t$ and $\mathcal{M}\_{t'}$ gives rise to non-stationarity even when those two time slices $\mathcal{G}\bigl[\mathbf{V}_t \bigr]$ and $\mathcal{G}\bigl[\mathbf{V}\_{t'} \bigr]$ are graphically identical. This is described in `Appendix E`.
> > > > - **(iii)** Regarding the term "active parents": we agree this could be misleading without clarification. We used it in the response to aid the reviewer’s intuition. What we meant is that the **set of actual causes** (i.e., variables that enter into the structural function $f_{V_{t'}}$) may expand over time because of transition edges. These parents become active in the sense that they influence the functional form of downstream variables. For example, in Figure 1(a), $X_t$ is not a parent of $Z\_{t'}$ in the intra-slice graph $\mathcal{G}\bigl[\mathbf{V}\_{t'} \bigr]$, but becomes a genuine (active) parent due to the transition edge $X_t \rightarrow Z\_{t'}$ in the time window $\mathcal{G}\bigl[\mathbf{V}_t \cup \mathbf{V}\_{t'}\bigr]$.
> > > >
> > > > In short, we can intuitively describe the origin of the non-stationarity like these steps below.
> > > >
> > > > **(1)** There are two different temporal models $\mathcal{M}_t$ and $\mathcal{M}\_{t'}$
> > > > where $\mathcal{M}_t = \langle \mathbf{U}_t, \mathbf{V}_t, \mathcal{F}_t, P(\mathbf{U}_t \mid \mathbf{u}^\star_t) \rangle$ and $\mathcal{M}\_{t'} = \langle \mathbf{U}\_{t'}, \mathbf{V}\_{t'}, \mathcal{F}\_{t'}, P(\mathbf{U}\_{t'} \mid \mathbf{u}^\star\_{t'}) \rangle$
> > > >
> > > > **(2)** If $f_{V_t}  \in \mathcal{F}_t \neq f\_{V\_{t'}}  \in \mathcal{F}\_{t'}$, then those two temporal models  $\mathcal{M}_t$ and $\mathcal{M}\_{t'}$ are not equal.
> > > >
> > > > **(3)** This situation causes $P^{\mathcal{M}_t}(Y_t \mid \operatorname{do}(\mathbf{x}_t)) \neq P^{\mathcal{M}\_{t'}}(Y\_{t'} \mid \operatorname{do}(\mathbf{x}\_{t'}))$ because the interventional distributions $P^{\mathcal{M}\_{t'}}(Y_t \mid \operatorname{do}(\mathbf{x}_t))$ and $P^{\mathcal{M}\_{t'}}(Y\_{t'} \mid \operatorname{do}(\mathbf{x}\_{t'}))$ are calculated under different temporal models $\mathcal{M}_t$ and $\mathcal{M}\_{t'}$, respectively. Note that the causal graphs $\mathcal{G}\bigl[\mathbf{V}_t \bigr]$and $\mathcal{G}\bigl[\mathbf{V}\_{t'} \bigr]$ of these two temporal models have identical structure under Assumption 3.2.
> > > >
> > > > The details are available in `Appendix E`. These evolving mechanisms yield different interventional distributions over time, even when the visible graph structure remains unchanged. We will revise the main body of our manuscript to clarify this point.
> > > >
> > > > We hope this clarification sheds light on our approach to non-stationarity. Please feel free to let us know if there are any remaining concerns—we would be happy to elaborate further.

---

### Author Response · Authors · 2025-08-09
**Discussion summary**

Dear Area Chair,

Thank you once again for facilitating the author-reviewer discussion. We also extend our gratitude to all the reviewers for dedicating their time and effort to discuss our work.

As the discussion period draws to a close, we would like to summarize the key points that arose during our interactions.

In summary, the rebuttals from reviewers primarily focused on common concerns: contributions beyond prior work (`4V3R`, `Jv4n`, `k47E`), assumptions (`KTWZ`, `EXe2`, `k47E`), and scalability and regret bound (`k47E`, `Jv4n`, `KTWZ`, `Jv4n`).

We clarified that our work extends beyond prior structural causal bandit frameworks by addressing a fundamentally more challenging setting of non-stationary causal structures evolving over time, requiring identification of sequences of interventions rather than single-step ones. To this end, we developed novel theoretical graphical tools (IB$^+$ and QIB), established key propositions and theorems, and designed a sound and complete recursive algorithm for enumerating all POMIS$^+$. These contributions collectively represent a significant departure from existing methods.

Regarding assumptions, we explained how our framework remains valid under relaxed assumptions, including more general temporal dependencies (i.e., n-th order cases under Assumption 3.1) and structural changes in the causal graph (i.e., Assumptions 3.2 and 3.3). These are described in `Appendix G` and `Appendix L`. We will explicitly clarify these points and their implications in the final manuscript to improve accessibility.

On scalability and regret, we explained that our algorithm serves as an offline, graph-based preprocessing step that prunes the intervention space before any online learning. This reduces computational burden—especially in settings with repeated or modular causal structures—and can indirectly improve regret by enabling downstream bandit algorithms to operate over a substantially smaller action set.

Reviewer `KTWZ` expressed satisfaction with our rebuttal and suggested that including some discussion from the response will enhance our manuscript, which we will certainly incorporate in our revision.

Reviewer `EXe2` raised a question regarding the source of non-stationarity. We addressed that our framework of non-stationarity is founded on the Temporal Model (Definition 3.2) and Assumptions 3.2 and 3.3.

Reviewer `4V3R` was largely convinced by our additional clarifications regarding the motivation and relevance of our framework. The reviewer also agreed that understanding complex causal structures is important for sensitive decision-making processes. We will definitely clarify our complex notations in the revision, following the reviewer’s suggestion.

After our responses to the rebuttals, both reviewers `Jv4n` and `k47E` commented on broader limitations of the structural causal bandit (SCB) framework. While our current approach focuses on offline pruning based on causal graph semantics, we agree that extending it beyond the offline phase could be an interesting direction for future work.

Throughout the discussion period, we believe we have successfully resolved all the concerns raised by the reviewers.

We thank the reviewers for their constructive feedback, which has helped us refine the presentation, clarify key assumptions, and better articulate the novelty and relevance of our contributions. We look forward to further advancing this line of research within the community.

Sincerely,

Authors of Submission 27658

---

### Note · Authors · 2025-08-13

Dear reviewers and AC,

We sincerely thank you for your dedicated time and insightful feedback on our paper.

During the rebuttal and discussion period, the main concerns raised by the reviewers focused on our assumptions, presentation of our contributions, and the scalability and regret bounds. Other valuable points included the source of non-stationarity and our complex notations.

We are grateful for these comments, as they have given us a clearer path to improve our work. We will thoroughly revise the manuscript to enhance its clarity and strengthen our arguments. We are confident that these constructive suggestions will improve the final version of our paper.

Specifically:

1. We will improve the presentation of our contributions in the introduction that our challenge lies in identifying **sequences of interventions** in non-stationary causal structures evolving over time. We will also enhance readability by clarifying notations throughout the manuscript.
2. Regarding the assumptions already stated in `Appendix G` and `Appendix L`, we will revise and clarify that our algorithm remains valid even without Assumptions 3.2 and 3.3, and applies to any n-th order cases under Assumption 3.1. We will include this point in the main body.
3. We will clarify the scalability and regret bounds. Since our algorithm functions as an offline, graph-based preprocessing step that prunes the intervention space prior to any online learning, the scalability and regret can be reduced in the presence of repeated causal structures in practice. Furthermore, we will emphasize that our proposed method allows the use of standard bandit algorithms to analyze regret.
4. We will explicitly state that the origin of non-stationarity stems from distributional shifts caused by the existence of transition edges, which alter temporal models even if the causal graph structure remains unchanged or varies.

We sincerely value all the comments provided. We assure you that we will address each point in our revision to enhance and strengthen the manuscript.

Sincerely,

Authors of Submission 27658

---

### Decision · Program_Chairs · 2025-09-17

**Decision:**

Accept (poster)

**Comment:**

The authors propose an extension of the POMIS framework to time-series data. They do this in the context of non-stationary causal bandit algorithms. The scores are mixed and the paper as it stands currently is borderline.

Applying POMIS on the time-unfolded graph does make sense. The technical contribution in this sense may be limited. The insight that comes with it is valuable: Intervene a priori and clamp down that intervention to identify the optimal arm later.

Some concerns of critical reviewers I do not agree with: Assumptions 3.1 and 3.2 seem sensible to me. I am pretty sure 3.1, especially, can be relaxed without many issues. I also am not concerned about practical utility as this is a fundamental contribution. The lack of regret bound is a valid concern. The causal structure certainly could have been used in other ways instead of only pruning the action space, but this is not grounds for rejection.